# Rethinking Evaluation of Sparse Autoencoders through the Representation of Polysemous Words

**Gouki Minegishi**[1]    **Hiroki Furuta**[1]    **Yusuke Iwasawa**[1]    **Yutaka Matsuo**[1]
[1]The University of Tokyo
`{minegishi,furuta,iwasawa,matsuo}@weblab.t.u-tokyo.ac.jp`

## Abstract

Sparse autoencoders (SAEs) have gained a lot of attention as a promising tool to improve the interpretability of large language models (LLMs) by mapping the complex superposition of *polysemantic* neurons into *monosemantic* features and composing a sparse dictionary of words. However, traditional performance metrics like Mean Squared Error and $L_0$ sparsity ignore the evaluation of the semantic representational power of SAEs — whether they can acquire interpretable monosemantic features while preserving the semantic relationship of words. For instance, it is not obvious whether a learned sparse feature could distinguish different meanings in one word. In this paper, we propose a suite of evaluations for SAEs to analyze the quality of monosemantic features by focusing on polysemous words. Our findings reveal that SAEs developed to improve the MSE-$L_0$ Pareto frontier may confuse interpretability, which does not necessarily enhance the extraction of monosemantic features. The analysis of SAEs with polysemous words can also figure out the internal mechanism of LLMs; deeper layers and the Attention module contribute to distinguishing polysemy in a word. Our semantics-focused evaluation offers new insights into the polysemy and the existing SAE objective and contributes to the development of more practical SAEs[1].

## 1 Introduction

Sparse autoencoders (SAEs) have garnered significant attention in the field of mechanistic interpretability research (Sharkey et al., 2022; Bricken et al., 2023). SAEs aim to address the *superposition hypothesis* (Elhage et al., 2022), which suggests that activations in large language models (LLMs) can become *polysemantic*[2] — that is, a single neuron corresponds to multiple functionalities — posing a significant challenge for interpretability. By learning to reconstruct LLM activations, SAEs address this challenge by mapping polysemantic neurons to an interpretable space of *monosemantic* features (Sharkey et al., 2022; Bricken et al., 2023). Recent studies have demonstrated the utility of SAEs in various contexts. Templeton et al. (2024) employed SAEs to observe features related to a broad range of safety concerns in Claude 3 Sonnet model, including deception, sycophancy, bias, and dangerous content. Similar investigations have been conducted on GPT-4 (Gao et al., 2024) and Gemma (Lieberum et al., 2024). Moreover, the application of SAEs has extended beyond LLMs to diffusion models (Daujotas, 2024) and reward models (Riggs & Brinkmann, 2024).

SAEs have an architecture consisting of a single, wide layer in the autoencoder, and their training typically involves a reconstruction loss coupled with an $L_1$ penalty on the features (Sharkey et al., 2022; Bricken et al., 2023). This approach aims to learn features that are both sparse and accurate, creating a known trade-off between Mean Squared Error (MSE) and $L_0$ sparsity. Previous research has focused on improving this MSE-$L_0$ Pareto frontier, leading to proposals such as Gated SAE, TopK, and JumpReLU (Rajamanoharan et al., 2024a;b; Gao et al., 2024). Despite improvements in

---

[1]Code: `https://github.com/gouki510/PS-Eval`, Dataset: `https://huggingface.co/datasets/gouki510/Wic_data_for_SAE-Eval`

[2]In this paper, we distinguish *polysemantic/monosemantic* as multiple/single functionalities in the activation of LLMs (Elhage et al., 2022), and *polysemous/monosemous* as multiple/single meanings in the words.

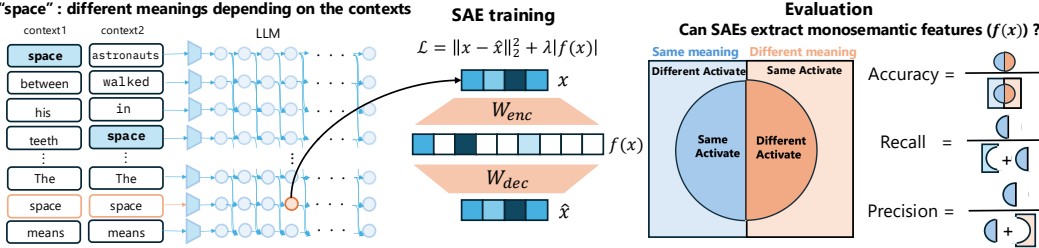

Figure 1: Evaluation of SAE's ability to extract monosemantic features from polysemantic activations in LLM. The word "space" in inputs tokens has different meanings depending on the context: context1 refers to "space" as in the universe (e.g., *...astronauts walked in space...*), while context2 refers to "space" as in a gap or physical distance (e.g., *...space between his teeth...*). SAE is trained by minimizing reconstruction loss and applying a sparsity constraint. The evaluation examines whether the features produced by the SAE can differentiate between the meanings of "space" in different contexts.

the MSE-$L_0$ Pareto frontier, it serves only as a substitute for the original goal of finding monosemantic features, and we do not evaluate it in practice. Such a gap highlights concerns about evaluation methodologies in SAEs, emphasizing the need for approaches that better align with the objective of discovering meaningful and interpretable features. Motivated by this, prior studies have utilized the IOI task (Wang et al., 2022) with known internal circuits in GPT-2 small (Cunningham et al., 2023; Makelov et al., 2024), RAVEL benchmark with counterfactual labels (Chaudhary & Geiger, 2024), and targeted probe perturbation (Karvonen et al., 2024b), as well as automated assessments using GPT-4 (Cunningham et al., 2023; Karvonen et al., 2024a). However, these metrics are limited to specific models or functional tasks and do not directly assess whether SAE features genuinely represent monosemantic characteristics, which was the original motivation for introducing SAEs.

To address these issues, we evaluate whether SAE features can effectively learn to decompose polysemantic LLM activations into monosemantic features by proposing a metric and dataset, Poly-Semantic Evaluations (PS-Eval). In natural language, polysemantic activations may naturally result from the polysemous words (Li et al., 2024). PS-Eval measures whether SAEs can correctly extract monosemantic features from polysemantic activations (Figure 1), based on the Word-in-Context (WiC) dataset (Pilehvar & Camacho-Collados, 2019), which contains polysemous words whose meanings vary based on context. PS-Eval can assess whether different features in SAE are activated when tokens with different meanings are input, and whether the same features in SAE are activated for tokens with the same meaning. Specifically, we construct a confusion matrix (Table 2) and evaluate the SAE using accuracy, precision, recall, and specificity metrics. To clarify whether our polysemous evaluation helps analyze polysemantic features in SAEs, we employed the logit lens (nostalgebraist, 2020) to examine the impact of monosemantic SAE features on subsequent layers of LLMs. Even in polysemous contexts, these SAE features represent distinct meanings, which indicates that SAEs extract monosemantic features that meaningfully affect the semantic processing in subsequent layers of LLMs.

We conducted experiments to determine which types of SAEs are more conducive to learning monosemantic features. We observed that increasing the dimensionality of SAE latent variables generally leads to better acquisition of monosemantic features, consistent with common scaling practices in SAE research. However, the accuracy saturates as the latent dimension increases, suggesting that there is an optimal latent dimension for practical use of SAE (Templeton et al., 2024; Gao et al., 2024). We also evaluated activation functions such as TopK and JumpReLU, which aim to improve the MSE-$L_0$ Pareto frontier. However, our results show that these functions do not perform well according to our F1 score metrics. Specifically, the performance ranking is ReLU > JumpReLU > TopK. This indicates that solely optimizing for MSE and $L_0$ does not evaluate the ability to extract monosemantic features, highlighting the strengths of our proposed evaluations.

Next, we investigate the specific locations in LLMs where SAEs can learn monosemantic features. We first examined how layer depth affects specificity, defined as whether SAE features activate differently for words with different meanings. Our analysis found that deeper layers generally have higher specificity, which indicates that the interpretation of polysemantic tokens occurs gradually as the layers deepen. Also, contrary to the traditional MSE metric, which shows that SAE performance

generally decreases with deeper layers, specificity scores from PS-Eval improve, which highlights that SAE performance cannot be fully assessed using MSE alone. We further evaluated different Transformer components, such as residual layers, MLPs, and self-attention; even within the same layer, the performance of SAEs differed. When examining specificity, attention shows higher values than Residual and MLP. This suggests that even within a single layer of the Transformer, the Attention mechanism contributes to distinguishing polysemous words.

In summary, our primal contributions are:

1. **Introduction of Poly-Semantic Evaluation (PS-Eval)**: We propose a new evaluation metric and dataset, PS-Eval, designed to assess whether SAEs can effectively extract monosemantic features from polysemantic activations. PS-Eval provides a model-independent evaluation metric for the semantic clarity of SAE's features.

2. **Comprehensive Evaluation of SAE**: Our findings suggest that SAEs developed to improve the MSE-$L_0$ Pareto frontier do not necessarily improve the extraction of monosemantic features. This suggests the need for more holistic evaluation protocols complement to traditional Pareto efficiency.

3. **Insights into Layer Depth and Transformer Components**: Our findings reveal that deeper layers achieve higher specificity and that the attention mechanism contributes to distinguishing polysemous words.

## 2 RELATED WORKS

**Sparse Autoencoder**   Individual neurons in neural networks are polysemantic, representing multiple concepts, and each concept is distributed across many neurons (Smolensky, 1988; Olah et al., 2020; Elhage et al., 2022; Gurnee et al., 2023). SAE is a widely used method that maps these polysemantic neurons to monosemantic high-dimensional neurons, improving the interpretability of activations in neural networks (Sharkey et al., 2022; Bricken et al., 2023). In practice, such polysemantic SAE features have been shown to correspond to diverse semantics, ranging from concrete meanings such as *Golden Gate Bridge* to more abstract concepts like the *entire Hebrew language* (Bricken et al., 2023; Templeton et al., 2024). Recently, SAEs have been applied to large language models such as Claude Sonnet (Templeton et al., 2024), GPT-4 (Gao et al., 2024), and Gemma (Lieberum et al., 2024), where they have been used to discover dangerous content and biased features, as well as to control the behavior of LLMs. SAEs are trained by reconstructing the activations of LLMs and an $L_1$ sparsity loss of SAE features, which introduces a trade-off between MSE (fidelity) and $L_0$ (sparsity). In previous research, to improve the MSE-$L_0$ Pareto frontier, the following approaches have been proposed: scaling the number of SAE features (Templeton et al., 2024; Gao et al., 2024), and introducing better activation functions, such as alternatives to ReLU (Rajamanoharan et al., 2024a; Gao et al., 2024; Rajamanoharan et al., 2024b).

One of the challenges with SAEs is the difficulty of evaluation. While MSE and $L_1$ sparsity are imposed during training, it is uncertain whether improving this Pareto optimality actually leads to the acquisition of monosemantic features. Existing research has used evaluations like the IOI task in GPT-2 (Makelov et al., 2024; Cunningham et al., 2023), where internal circuits are well understood, or automated evaluations using GPT-4 (Cunningham et al., 2023; Karvonen et al., 2024a). However, there is currently no direct evaluation metric to assess whether SAEs achieve their primary goal of mapping polysemantic activations into monosemantic spaces. In this study, we evaluate how well SAEs can acquire monosemantic representations at the level of individual word meanings using the Word in Context dataset (Pilehvar & Camacho-Collados, 2019), which includes polysemous tokens with meanings that vary depending on context.

**Context Sensitive Representations**   In natural language processing, word embeddings have been limited by their ability to represent only static features, where each word is associated with a single embedding regardless of the context in which it appears. To address this limitation, the Word-in-Context (WiC) dataset (Pilehvar & Camacho-Collados, 2019) was proposed for the evaluation of context-sensitive word embeddings. Currently, a multilingual version of WiC (Raganato et al., 2020) has been developed, and it has also become a part of the widely-used SuperGLUE benchmark (Wang et al., 2020). Because polysemantic features may naturally come from the polysemous words (Li

et al., 2024), we evaluate whether SAEs can learn monosemantic features by using datasets with polysemous words – where the same token has different meanings – such as WiC.

## 3 PRELIMINARIES

We here summarize the key concepts and notation required to understand SAE architectures and their training procedures. The central goal of SAEs during training is to decompose a model's activation $\mathbf{x} \in \mathbb{R}^n$ into a sparse, linear combination of feature directions, expressed as:

$$\mathbf{x} \approx \mathbf{x_0} + \sum_{i=1}^{M} f_i(\mathbf{x})\mathbf{d}_i, \tag{1}$$

where $\mathbf{d}_i$ represents unit-norm latent feature directions ($M \gg n$), and the sparse coefficients $f_i(\mathbf{x}) \geq 0$ correspond to feature activations for the input $\mathbf{x}$. This framework aligns with the structure of an autoencoder, where the input $\mathbf{x}$ is encoded into a sparse vector $\mathbf{f}(\mathbf{x}) \in \mathbb{R}^M$ and decoded to approximate the original input.

### 3.1 ARCHITECTURE

The typical SAE is structured as a single-layer autoencoder, where the encoding and decoding functions are defined as follows:

$$\mathbf{f}(\mathbf{x}) := \sigma(W_{\text{enc}}(\mathbf{x} - \mathbf{b}_{\text{dec}}) + \mathbf{b}_{\text{enc}}), \tag{2}$$

$$\hat{\mathbf{x}}(\mathbf{f}(\mathbf{x})) := W_{\text{dec}}\mathbf{f}(\mathbf{x}) + \mathbf{b}_{\text{dec}}. \tag{3}$$

Here, $W_{\text{enc}} \in \mathbb{R}^{Rn \times n}$ and $W_{\text{dec}} \in \mathbb{R}^{n \times Rn}$ are the encoding and decoding weight matrices, respectively, where $R$ is the expand ratio that determines the dimension of the intermediate in SAE. The vectors $\mathbf{b}_{\text{enc}}$ and $\mathbf{b}_{\text{dec}}$ represent the biases. We initialize the $W_{\text{dec}}$ as the transposed $W_{\text{enc}}$, following the methodology from previous studies (Gao et al., 2024). The sparse encoding $\mathbf{f}(\mathbf{x}) \in \mathbb{R}^{Rn}$ serves as the intermediate representation, while $\hat{\mathbf{x}}(\mathbf{f})$ reconstructs the input $\mathbf{x}$ from the sparse encoding. The columns of $W_{\text{dec}}$ correspond to the learned feature directions $\mathbf{d}_i$, and the elements of $\mathbf{f}(\mathbf{x})$ represent the activations for these directions. $\sigma(\cdot)$ is an activation function; we adopt ReLU, TopK (Gao et al., 2024), or JumpReLU, including its STE variant (Rajamanoharan et al., 2024b) in this paper, while using ReLU by default.

### 3.2 TRAINING METHODOLOGY

Training SAE involves optimizing a loss function that balances reconstruction accuracy and sparsity. The reconstruction loss is defined as the squared error between the input $\mathbf{x}$ and the reconstruction $\hat{\mathbf{x}}(\mathbf{f}(\mathbf{x}))$ and sparsity in the activations $\mathbf{f}(\mathbf{x})$ is encouraged using the $L_1$ norm. The overall loss function is a combination of these two terms, weighted by a sparsity regularization factor $\lambda$:

$$\mathcal{L}_{\text{SAE}}(\mathbf{x}) = \mathcal{L}_{\text{recon}}(\mathbf{x}) + \lambda\mathcal{L}_{\text{sparse}}(\mathbf{f}(\mathbf{x})) = \|\mathbf{x} - \hat{\mathbf{x}}(\mathbf{f}(\mathbf{x}))\|_2^2 + \lambda\|\mathbf{f}(\mathbf{x})\|_1. \tag{4}$$

This formulation encourages the autoencoder to reconstruct the input faithfully while enforcing sparsity in the latent features. To prevent trivial solutions, such as reducing the sparsity penalty by scaling down the encoder outputs and increasing the decoder weights, it is necessary to constrain the norms of the columns of $W_{\text{dec}}$ during training. In addition, we incorporate an auxiliary loss known as Ghost Grads (Jermyn & Templeton, 2024), which reduces the occurrence of dead latents where SAE features stop activating for long periods. In Appendix A, we discusses the relationship between dead latents and Ghost Grads.

Following prior work (Templeton et al., 2024), we use an expand ratio of $R = 32$ and a sparsity regularization factor of $\lambda = 0.05$ by default for training SAE. The base LLM used as activations for the SAE is GPT-2 small (Radford et al., 2019). Unless specified otherwise, activations are extracted from the 4th layer. Inspired by prior work (Makelov et al., 2024), the training data consisted of the WiC dataset (Raganato et al., 2020) for in-domain data (i.e., data from the same domain as the evaluation dataset) and RedPajama (Weber et al., 2024) for open-domain data (i.e., data from a general domain used during the pre-training of LLMs). The distinction between open-domain and in-domain data is further explained in Appendix B. By default, we use in-domain data for training. Additional training details, such as hyper-parameters, are provided in Appendix C.

Table 1: Statistics and examples of `poly-contexts` and `mono-contexts` in PS-Eval.

| # of Total Samples | 1112 (label 0: 556, label 1: 556) |
|---|---|
| **Example Sentences** | `poly-contexts` (**space**)
Context 1: *The astronauts walked in outer "space" without a tether.*
Context 2: *The "space" between his teeth.*
Label: 0 (**Different**)

`mono-contexts` (**space**)
Context 1: *They stopped at an open "space" in the jungle.*
Context 2: *The "space" between his teeth.*
Label: 1 (**Same**) |

## 3.3 TRADE-OFF BETWEEN RECONSTRUCTION AND SPARSITY

As shown in Equation 4, a critical aspect of evaluating SAE during training lies in balancing two primary metrics: mean squared error for reconstruction accuracy and the $L_0$ norm for measuring sparsity. These metrics often exhibit a trade-off achieving lower MSE typically results in less sparsity, and increasing sparsity generally raises the reconstruction error. This trade-off can be represented by the Pareto frontier, as shown in Figure 2, where the x-axis corresponds to the $L_0$ norm (sparsity) and the y-axis represents MSE (reconstruction error). Ideally, optimal models occupy the lower-left part of the figure, where both MSE and sparsity are minimized simultaneously.

To improve the Pareto frontier, there are two main strategies: increasing the number of latent variables and enhancing activation function. For increasing latent variables, scaling laws have shown that higher dimensionality improve this Pareto frontier.

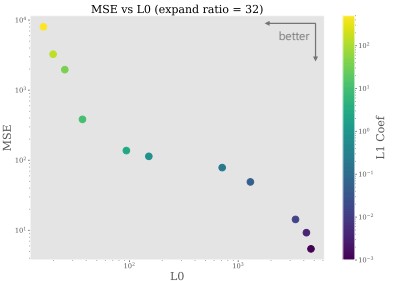

Figure 2: MSE v.s. $L_0$ for expand ratio = 32 across varying $L_1$ coefficients ($\lambda$). The color gradient represents the $L_1$ coefficient values, with brighter colors indicating larger $\lambda$. As the $\lambda$ increases, the model becomes sparser (lower $L_0$) while MSE gets worse.

## 4 PS-EVAL

As a holistic evaluation of SAE considering the semantics in the features, we introduce Poly-Semantic Evaluation (PS-Eval), a suite of datasets and metrics for evaluation. PS-Eval consists of polysemous words whose meanings are different based on the given context, which aims to evaluate whether SAE can map entangled polysemantic neurons in LLMs into simple monosemantic features for better interpretability while distinguishing the different meanings.

## 4.1 DATASET

We build PS-Eval on top of the WiC dataset (Pilehvar & Camacho-Collados, 2019), which is a rich source of polysemous words. The original WiC dataset provides two contexts with a shared target word for each sample, and its task is to determine whether the target word holds the same meaning or a different meaning in both contexts. The dataset has binary labels, where a label of **0** indicates that the target word has **different** meanings in the two contexts, and a label of **1** signifies that the target word's meaning remains the **same** in both context. For PS-Eval, we carefully select instances from the WiC dataset whose target word is tokenized as a single token in GPT-2 small.

We refer to the two contexts with a label of 0 (indicating different meanings) as `poly-contexts` (**target word**). Table 1 shows an example using the target word "space" in poly-context. Even though the tokenization of "space" remains the same in both contexts, the word carries different meanings: in the first context, it refers to "universe" while in the second, it refers to "an area or gap". On the other hand, when the label is 1, the meaning of the target word remains the same across both contexts. We denote these as `mono-contexts` (**target word**). Table 1 also has

Table 2: (**Left**) Confusion matrix for polysemy detection in SAE evaluation. Note that *Positive* or *Negative* labels are just for convenience to describe the confusion matrix, and these do not mean that, it is bad to identify that the different words have different features or it is good to identify that the same words have the same features. (**Right**) metrics (accuracy, recall, precision, and specificity) and their interpretations.

|  | **Mono-context** | **Poly-context** |
|---|---|---|
| Same Max Activated Feature | Same meaning, same feature (**T**rue **P**ositive) | Different meaning, same feature (**F**alse **P**ositive) |
| Different Max Activated Feature | Same meaning, different feature (**F**alse **N**egative) | Different meaning, different feature (**T**rue **N**egative) |

| Metric | Formula | Interpretation |
|---|---|---|
| Accuracy | $\frac{TP+TN}{TP+FP+TN+FN}$ | How well the model predicts the overall situation |
| Recall | $\frac{TP}{TP+FN}$ | When the same meaning's words are given, how often they are identified as the same |
| Precision | $\frac{TP}{TP+FP}$ | When the same features are activated, how often the meanings are same |
| F1 | $\frac{2\times Recall\times Precision}{Recall+Precision}$ | How Recall and Precision are balanced |
| Specificity | $\frac{TN}{TN+FP}$ | When the different meaning's words are given, how often they are identified as different |

an example using the target word "space" in mono-contexts. Here, the word "space" refers to "an area or gap" in both contexts, and thus the label is 1, signifying that the meaning is consistent.

## 4.2 EVALUATION METRICS

PS-Eval focuses on analyzing activations of the SAE for the target word across different contexts to determine whether SAE can extract monosemantic features.To identify the activation related to polysemous target words, we consider the following formatted input:

$$\{\texttt{context}\}.\ \textit{The}\ \{\texttt{target word}\}\ \textit{means}$$

where the $\{\texttt{context}\}$ is an example sentence of $\{\texttt{target word}\}$, reflecting either polysemous contexts (poly-context) or monosemous contexts (mono-context). As done in Wang et al. (2022) such a format sentence makes it easy to identify the corresponding activation. In this paper, the activation of the token for the $\{\texttt{target word}\}$ at layer $l$ is denoted as $a^l_{\text{LLM}}(\text{context})$, and the feature from the SAE is represented as $f^l_{\text{SAE}}(\text{context})$. Figure 1 (left) describe the conceptual workflow.

**Confusion Matrix for Polysemy Detection** To evaluate the ability of SAEs to extract monosemantic features, we construct a confusion matrix (Table 2; left) based on the maximum activations of the target word in different contexts. Each cell in the matrix is defined as follows:

- In mono-context, where the target word has the same meaning in both contexts, we expect the same feature with the highest activation to appear. This corresponds to a True Positive (TP).
- In mono-context, if different features with the highest activation appear, this results in a False Negative (FN).
- In poly-context, where the target word has different meanings, we expect different features with the highest activation to appear, representing a True Negative (TN).
- In poly-context, if the same feature with the highest activation appears for the target word, this results in a False Positive (FP).

Note that *positive* or *negative* labels here are just for convenience to describe the confusion matrix, and these do not mean that, it is undesirable to identify that the different words have different features or it is desirable to identify that the same words have the same features. After classifying each instance into a confusion matrix, we then calculate the quantitative metrics such as **accuracy**, **precision**, **recall**, **F1 score** and **specificity** to assess whether the SAE is capable of acquiring monosemantic features (Table 2; right); Recall measures how often the same meaning words have the same SAE features. Precision measures how often the same SAE features are attributed to the same meaning words. Accuracy and F1 score can quantify the overall performance. Specificity measures how often the **different** meaning words have the **different** SAE features.

If SAEs have ideal features, all of these metrics can be high simultaneously. We advocate for a comprehensive evaluation that considers all metrics holistically, in complement to existing performance measures, such as MSE, $L_0$, or IOI performances. While measuring individual metrics may also be useful to improve SAEs, we note that we need to carefully treat Specificity when solely interpreting it. Because Specificity measures when the different meaning's words are given, how often they are identified as different, it may reach an unfairly high value by definition if SAE features have significantly different representations of each other. For comparison, we investigate the performance of an SAE that activates *randomly* (see Appendix D) and a *dense* SAE that is trained without $L_1$

Table 3: Comparison of Top-7 logits in `poly-context` (space), `poly-context` (save) and `poly-context` (ball). The synonyms or words relevant to the meaning of context 1 and context 2 are coloring. We also highlight the splitted token (e.g. ball-oons) in GPT-2 Small. The words that appear in both context 1 and 2 are **bolded**.

| | poly-context (**space**) | | | | poly-context (**save**) | | | | poly-context (**ball**) | | | |
| | Context 1 | | Context 2 | | Context 1 | | Context 2 | | Context 1 | | Context 2 | |
| rank | tokens | logits | tokens | logits | tokens | logits | tokens | logits | tokens | logits | tokens | logits |
|---|---|---|---|---|---|---|---|---|---|---|---|---|
| 1 | flight | 1.224 | Layout | 0.894 | **Save** | 0.867 | eLL | 0.874 | oons | 1.170 | handler | 0.997 |
| 2 | plane | 0.957 | occupied | 0.882 | ulative | 0.696 | Cache | 0.744 | istics | 1.115 | player | 0.884 |
| 3 | **shuttle** | 0.938 | spaces | 0.853 | luc | 0.681 | TOR | 0.731 | oon | 1.026 | ball | 0.852 |
| 4 | gravity | 0.937 | vacated | 0.846 | Saving | 0.675 | Marcos | 0.665 | antine | 1.008 | wright | 0.842 |
| 5 | craft | 0.920 | space | 0.825 | entary | 0.672 | Sanctuary | 0.658 | park | 0.975 | hand | 0.832 |
| 6 | Engineers | 0.876 | **shuttle** | 0.799 | WhatsApp | 0.655 | **Save** | 0.649 | asso | 0.960 | players | 0.825 |
| 7 | planes | 0.869 | occupancy | 0.798 | saving | 0.650 | orage | 0.644 | asted | 0.905 | disposal | 0.814 |

regularization (see Appendix E). These results show that random SAE achieves the upper bound of Specificity, while densely trained SAE does not exhibit such unnatural metrics. Therefore, we basically recommend a holistic evaluation with all metrics, and, in case you would like to optimize one of them independently, we recommend the usage of class-balanced metrics such as accuracy or F1 scores as safer options.

## 5 MONOSEMANTIC FEATURE MAY CAPTURE EACH MEANING IN POLYSEMY

PS-Eval is designed to evaluate SAEs through the analysis of whether the learned monosemantic features encoded from the polysemous words correspond to each word meaning, depending on the context. To clarify this core concept, we start with validating our implicit assumption of whether the SAE features encode word meanings by employing the logit lens technique (nostalgebraist, 2020).

We compute the maximum activated SAE features when two poly-context samples (Context 1, Context 2) are input. The corresponding activations for each context are calculated as:

$$a_{\max}^1 = W_{\text{dec}} \max(f(x_1)), \quad a_{\max}^2 = W_{\text{dec}} \max(f(x_2)),$$

where $x_1$ and $x_2$ represent the activations of the LLM for Context 1 and Context 2 in poly-context, respectively. Next, we apply the unembedding matrix to these activations to obtain the logits:

$$\text{logit}_1 = W_U a_{\max}^1, \quad \text{logit}_2 = W_U a_{\max}^2.$$

Table 3 presents the top-7 logits for the polysemous words "space", "save", and "ball" based on the above calculations. These contexts can be found in Table 6 in Appendix F. For Context 1 of the word "space", the top logits are related to terms associated with outer space (e.g., "flight", "gravity"), whereas for Context 2, the top logits are more aligned with architectural or layout-related terms (e.g., "Layout", "vacated"). This shows that the SAE features correspond to each word meaning. Furthermore, it can be observed that when the word meanings in the context are different, SAE features also reflect those distinct meanings. For the word "save," Context 1 focuses on terms related to saving in the sense of preserving or conserving resources, such as money (e.g., "Save," "Saving"), while Context 2 extracts meanings more related to saving data or technical operations (e.g., "Cache"). This distinction is clearly seen in Table 6, where Context 1 involves monetary or resource-saving, and Context 2 relates to data saving in a technical sense. Similarly, for the word "ball," Context 1 refers to a formal event or dance, where "ball" is used in the sense of a social gathering (e.g., "park"), while Context 2 reflects meanings associated with sports, where "ball" refers to a physical object used in games or sports activities (e.g., "handler," "player").

Following positive observations in prior works (Templeton et al., 2024; Cunningham et al., 2023; Li et al., 2024), these qualitative results demonstrate that a monosemantic SAE feature can correspond to a specific word meaning, which justifies our evaluation protocol.

## 6 EVALUATING SAEs WITH POLYSEMOUS WORDS

### 6.1 SAE FEATURES ARE MORE INTERPRETABLE THAN LLM ACTIVATIONS

To investigate the significance of using SAE for PS-Eval, we demonstrate that using SAE features results in better interpretability of polysemous contexts than the activations of LLMs. When a poly-

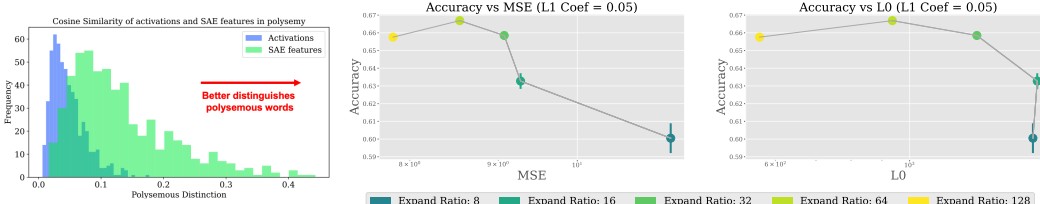

Figure 3: (**Left**) Cosine distance between LLM activations and SAE features in polysemantic contexts. The histogram compares the distribution of Polysemous Distinction (i.e. 1 - Cosine Similarity) for LLM activations (blue) and SAE features (green), demonstrating that SAE features extract more distinct representations in polysemous contexts. We adopt $R = 128$ and $L_1$ coefficient $\lambda = 0.05$ for the hyperparameters of SAEs. (**Right**) Comparison of Accuracy vs MSE and $L_0$ for different Expand Ratios ($L_1$ Coef = 0.05). The left panel shows the relationship between accuracy and mean squared error (MSE), while the right panel presents accuracy versus normalized $L_0$ sparsity. The markers represent various expand ratios (8, 16, 32, 64, and 128). Higher expand ratios generally show a trend towards better performance, as indicated by the yellow and green markers corresponding to expand ratios of 64 and 128.

semous context is input, we denote the LLM's activation for the target word as $a_{\text{LLM}}(\text{poly-context})$, and the SAE feature for the same context as $f_{\text{SAE}}(\text{poly-context})$. We calculated the cosine distance between these activations and SAE features to assess their similarity or divergence, which we refer to as *Polysemous Distinction*:

$$1 - \frac{a_{\text{LLM}}(\text{poly-context}_1) \cdot a_{\text{LLM}}(\text{poly-context}_2)}{\|a_{\text{LLM}}(\text{poly-context}_1)\|\|a_{\text{LLM}}(\text{poly-context}_2)\|}, \ 1 - \frac{f_{\text{SAE}}(\text{poly-context}_1) \cdot f_{\text{SAE}}(\text{poly-context}_2)}{\|f_{\text{SAE}}(\text{poly-context}_1)\|\|f_{\text{SAE}}(\text{poly-context}_2)\|}.$$

As shown in Figure 3 (right), the cosine distance between the LLM activations for different polysemous contexts is small, indicating that the LLM struggles to differentiate between distinct meanings in polysemous contexts (i.e., less interpretable activations). In contrast, SAE features show a larger cosine distance, effectively capturing the differences between the contexts with different meanings. This result demonstrates that SAE effectively disentangles different meanings, and, for polysemous words, it enhances the interpretability of internal representations in LLMs.

## 6.2 INCREASING EXPAND FACTOR IMPROVES POLYSEMOUS SAE FEATURE

Previous works (Templeton et al., 2024; Gao et al., 2024) have reported that increasing the dimensionality of SAE features leads to better performance in terms of MSE and $L_0$ metrics, and we may also expect that it allows for learning more fine-grained features from activations (Bricken et al., 2023). To investigate the relationship between PS-Eval performance and the scaling of latent variables, we evaluated the performance of our model trained with various expansion ratios (defined in Section 3.1) in Figure 3 (right). Similar to MSE and $L_0$, we observed that increasing the expand ratio led to higher accuracy on PS-Eval. This trend indicates that a higher expand ratio enables the model to better extract the monosemantic fetures from polysemantic activations. However, we found that the accuracy saturates around an expand ratio of 64, suggesting that there is an optimal latent dimension for practical SAE usage. Here, we define $L_0$ as the number of activated SAE features. See Appendix G for other metrics.

## 6.3 ALTERNATIVE ACTIVATION FUNCTIONS TO RELU ARE NOT ALWAYS BETTER

Various activation functions for SAE have been proposed to achieve a better Pareto optimality between MSE and $L_0$ sparsity. More recent work has introduced TopK, which replaces $\text{ReLU}(x)$ with $\text{TopK}(\text{ReLU}(x))$ (Gao et al., 2024), and JumpReLU, which modifies the threshold of the ReLU activation to improve performance in terms of both MSE and sparsity (Rajamanoharan et al., 2024b) (see Appendix H for the detailed descriptions).

We here evaluate whether these alternative activation functions also yield better results when applied to our evaluation metrics. In Figure 4, we compare models trained with TopK activations, where $k$ is varied over values such as $k = \{384, 192, 96\}$, and models trained with JumpReLU, including its STE variant. For JumpReLU, the threshold is adjusted over the range of $[0.0001, 0.001]$, while for the STE variant, the threshold represents an initial value. We note that TopK typically uses half the dimension of the model as the value of $k$ (e.g., for GPT-2 small, this corresponds to $768/2 =$

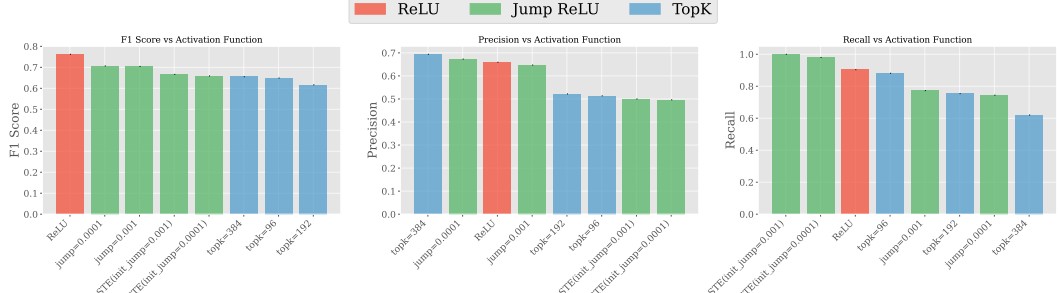

Figure 4: Comparison of F1 score, Precision, and Recall across different activation functions. The models were trained with JumpReLU, including its STE variant (with varying thresholds), TopK (with different $k$), and ReLU. The results imply that ReLU may be still better than JumpReLU or TopK with the polysemous words.

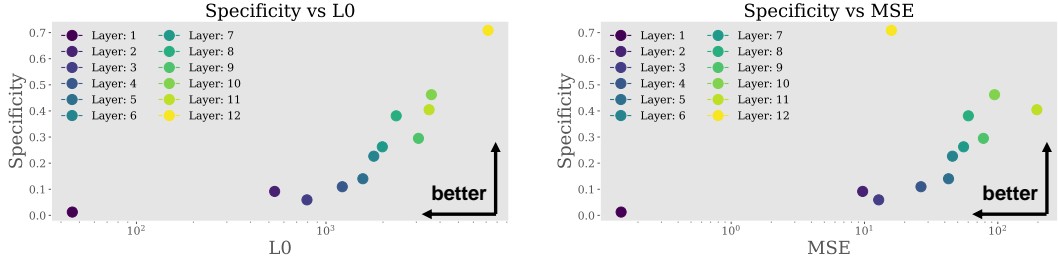

Figure 5: Comparison of Specificity across different layers of the SAE in relation to $L_0$ (left) and MSE (right). Each point represents a layer, from layer 1 to layer 12, with deeper layers (i.e., layer 6 onwards) showing an increasing trend in Specificity. This indicates that deeper layers exhibit more distinct activations for polysemous words, consistent with previous findings that deeper layers in SAEs face greater challenges in training, leading to higher MSE and $L_0$ scores.

384), and that JumpReLU often adopts a threshold of $0.001$ in the original works (Gao et al., 2024; Rajamanoharan et al., 2024b).

Figure 4 compares these activation functions in terms of F1 score, Precision, and Recall. ReLU achieves the highest F1 score, with the order being ReLU > JumpReLU > TopK. For TopK, smaller $k$ leads to higher Recall, and JumpReLU-STE further boosts Recall. These results suggest that focusing solely on the trade-off between MSE and $L_0$ may not effectively enhance semantic quality.

## 7 LOCATING POLYSEMY CIRCUITS IN LLMS

**Polysemy is Captured in Deeper Layers** We investigate where large language models (LLMs) interpret polysemous words differently by using Specificity (calculated as Specificity $= \frac{TN}{TN+FP}$) based on the confusion matrix from PS-Eval (see Table 2). Specificity allows us to assess whether the features in the SAE activate differently when presented with Poly-context (contexts involving words with defferent meanings).

Figure 5 shows the Specificity of each SAE layer in relation to $L_0$ and MSE. Previous studies (Gao et al., 2024) have demonstrated that training SAEs becomes increasingly difficult as the depth of the layers increases, which is consistent with our findings, as both MSE and $L_0$ rise in deeper layers. While these two metrics increase, we observe that Specificity improves around the 6th layer, indicating that polysemous words are more easily decomposed into monosemantic features through SAE training. These results demonstrate an inverse relationship between MSE/$L_0$ and Specificity, suggesting that focusing solely on improving the Pareto frontier of MSE or $L_0$, as emphasized in prior research, may overlook important insights.

**SAE Features Vary by Transformer Component** While early prior work (Bricken et al., 2023) primarily centered on the activations within MLP layers, more recent studies have broadened this scope not only to MLP layers but also to residual and attention layers (Kissane et al., 2024; He et al., 2024). We explore which components of a single Transformer layer contribute to the acquisition of monosemantic features from polysemous words. Specifically, we divided the activations from a single Transformer layer into three distinct outputs: Residual, MLP, and Attention. See Appendix I

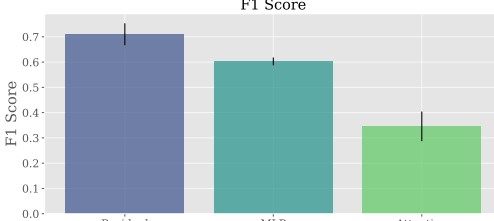 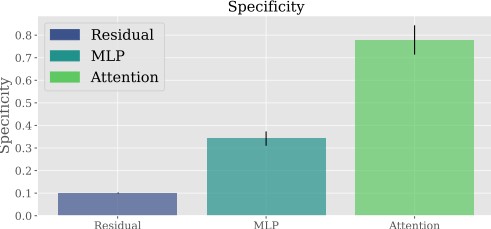

Figure 6: Comparison of F1 score and Specificity for SAEs trained on the outputs of the Residual, MLP, and Attention components. The left panel shows the F1 score, where Residual and MLP achieve higher values, while the Attention output performs lower. In contrast, the right panel shows Specificity, where the Attention output exhibits a significantly higher Specificity, despite the lower F1 score. See Appendix I for details on these components.

for details on these components. We trained SAEs on each of these components under consistent conditions, employing an $L_1$ regularization coefficient of 0.05, an expand ratio of 32, and focusing on the fourth layer of GPT-2.

From Figure 6, it is evident that while the Residual and MLP components achieve higher F1 scores, the Attention output lags behind. This suggests that, based on the harmonic mean of precision and recall, the features learned by the SAE from the Attention output do not activate examples with the same meaning as effectively as those learned from the Residual and MLP outputs. However, when examining Specificity, the Attention output demonstrates a markedly higher value compared to the Residual and MLP components. This indicates that the Attention mechanism is more effective at distinguishing between different features, specifically in identifying the different words with different features.

## 8 DISCUSSION AND LIMITATION

One potential limitation of PS-Eval lies in its reliance on evaluating the maximum activation of SAE features, which may introduce edge cases and failure modes. As discussed in Section 4.2 and illustrated in Figure 1 (left), PS-Eval is designed to measure how well SAE features respond to LLM activations associated with the meanings of target words. However, it is not guaranteed that the feature with the **maximum activation** always aligns with the intended word meanings. For instance, as observed in the logit lens results presented in Table 3, while SAE features often exhibit strong activation for specific word meanings, this behavior does not consistently generalize across all samples or LLMs. In some cases, the feature with the highest activation may reflect unrelated aspects rather than the meaning of the target word, potentially introducing ambiguity into the evaluation. To make the evaluation more robust, it would be interesting and important future direction to incorporate similarity metrics that utilize the distribution of SAE features, rather than relying solely on maximum activation.

## 9 CONCLUSION

We propose Poly-Semantic Evaluation (PS-Eval), a new metric and dataset to assess the ability of SAEs to extract monosemantic features from polysemantic activations, in complement to traditional MSE or $L_0$ metrics. Our findings reveal that increased latent dimensions enhance Specificity up to a limit, while MSE-$L_0$-optimized activation functions do not always improve semantic extraction. Deeper layers and Attention mechanisms significantly boost specificity, underscoring the utility of PS-Eval. We hope our work encourages the community to conduct a holistic evaluation of SAEs.

ACKNOWLEDGMENTS

HF was supported by JSPS KAKENHI Grant Number JP22J21582.

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

APPENDIX

## A  GHOST GRAD

In this section, we describe the auxiliary loss, which serves as an additional regularization mechanism for latent representations during training. The goal is to reduce the likelihood of inactive latent units by penalizing reconstruction errors from a subset of dead latents. This approach is conceptually similar to the "ghost gradients" method (Jermyn & Templeton, 2024), though modified to handle high-dimensional latent spaces with a more targeted selection of inactive units.

During training, each latent variable is monitored for activation. Latents that have not been activated for a pre-specified number of tokens (typically 10 million tokens) are flagged as "dead." These latents are indexed by the parameter $k_{\text{aux}}$, which represents the top-$k$ latent dimensions with the least activation (usually set to $k_{\text{aux}} = 512$).

Given the reconstruction error $e = x - \hat{x}$, where $x$ is the input and $\hat{x}$ is the model's reconstructed output, we define the auxiliary loss $\mathcal{L}_{\text{aux}}$ as the squared error between the true reconstruction and the reconstruction obtained using only the dead latents. This is formulated as:

$$\mathcal{L}_{\text{aux}} = \|e - \hat{e}\|_2^2,$$

where $\hat{e}$ represents the reconstruction from the dead latents, computed as $\hat{e} = W_{\text{dec}}z$, where $W_{\text{dec}}$ is the decoder matrix and $z$ is the latent vector associated with the top-$k$ dead units.

The full loss during training is given by the sum of the main loss $\mathcal{L}_{\text{SAE}}$ (Equation 4) and the auxiliary loss, weighted by a small coefficient $\alpha$:

$$\mathcal{L}_{\text{total}} = \mathcal{L}_{\text{SAE}} + \alpha\mathcal{L}_{\text{aux}}.$$

In our experiments, $\alpha$ is typically set to 1/32, which balances the influence of the auxiliary term without dominating the primary loss. Importantly, since the encoder forward pass is shared between the primary and auxiliary losses, the computational overhead from adding this regularization is minimal, increasing overall cost by approximately 10%.

To investigate the impact of Ghost Grad, we tracked the number of dead latents in the SAE during training. Dead latents are defined as features that do not activate even once during 1000 training steps, following Gao et al. (2024). Figure 7 clearly show that introducing Ghost Grad significantly prevents the increase in dead latents. Additionally, the PS-eval evaluation also demonstrated better performance when Ghost Grad was applied. Therefore, in our paper, we include Ghost Grad as the default setting.

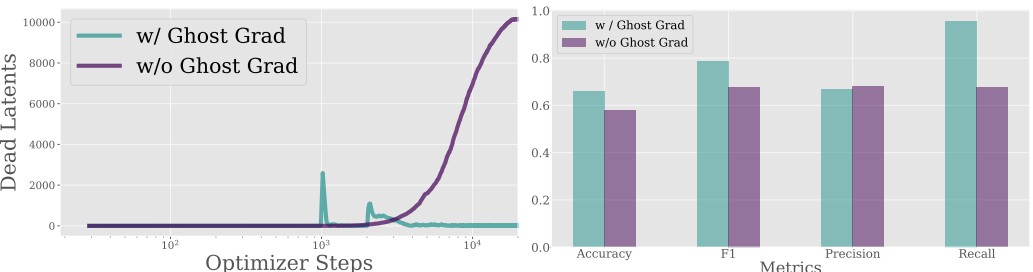

Figure 7: (**Left**) Changes in the number of dead features during training. The configuration used is expand ratio = 32, $L_1$ coefficient = 0.05 and layer 4. Without Ghost Grad, the number of dead features increases significantly, while using Ghost Grad prevents this growth. (**Right**) Relationship between the presence of Ghost Grad and evaluation metrics, showing differences in Accuracy, F1 score, Precision, and Recall. Using Ghost Grad consistently yields better results compared to not using it.

## B    OPEN DOMAIN VS IN-DOMAIN

As explained in prior work (Makelov et al., 2024), it remains unclear whether the dataset used for training the SAE should come from the same domain as the evaluation data (In Domain) or from a general dataset (Open Domain). In this section, we evaluate the SAEs trained on the WiC dataset (Pilehvar & Camacho-Collados, 2019) as the In Domain dataset and Red Pajama (Weber et al., 2024) as the Open Domain dataset. Figure 8 shows the results, where recall is higher for the Open Domain across all expand ratios, while precision is higher for models trained on the In Domain dataset.



Figure 8: Comparison of F1 Score, Recall, and Precision across different Expand Ratios (32, 64, 128) for Open Domain (red pajama (Weber et al., 2024)) and In Domain (WiC data) settings. The results show that Open Domain outperforms In Domain in terms of recall, while In Domain performs better in terms of precision.

## C    DETAILS OF EXPERIMENT

Here are the detailed parameters for this experiment. We initialize the weights of the Decoder as the transposed weights of the Encoder, following the methodology from previous studies (Gao et al., 2024).

Table 4: Summary of Training Configuration and Hyper-parameters.

| Parameter | Value |
|---|---|
| **Batch Size** | 8192 |
| **Total Training Steps** | 200,000 |
| **Learning Rate** | 2e-4 |
| **Input Dimension ($d_{\mathbf{in}}$)** | 768 (GPT2-small) |
| **Context Size** | 256 |

## D    RANDOM SAE

In this section, we analyze the performance of PS-Eval when SAEs are activated randomly. The experimental setup assumes an expand ratio of 32, meaning that the number of features in the SAE is calculated as expand ratio $\times d_{\mathrm{model}=32\times768=23576}$. We compute the probability of overlapping activations ($P_{\mathrm{same}}$) and the probability of non-overlapping activations ($P_{\mathrm{diff}}$).

The total number of ways to choose 2 features from 24576 features is given by:

$$\binom{24576}{2} = \frac{24576 \times (24576 - 1)}{2}$$

The probability of selecting the **same** feature is:

$$P_{\mathrm{same}} = \frac{1}{\frac{24576 \times (24576-1)}{2}} = \frac{2}{24576 \times 24575} = 3.3115039e - 9$$

The probability of selecting the **different** feature is:

$$P_{\mathrm{diff}} = 1 - P_{\mathrm{same}} = 0.99999999668$$

Next, we calculate the True Positives (TP), False Negatives (FN), False Positives (FP), and True Negatives (TN). We assume that the total number of data samples is $N = 1112$, and the number of samples with the same meaning is $n = 556$.

- True Positives (TP):
$$\text{TP} = n \times P_{\text{same}} \approx 0.00000184119$$

- False Negatives (FN):
$$\text{FN} = n \times P_{\text{diff}} \approx 555.999998154$$

- False Positives (FP):
$$\text{FP} = (N - n) \times P_{\text{same}} \approx 0.00000184119$$

- True Negatives (TN):
$$\text{TN} = (N - n) \times P_{\text{diff}} \approx 555.999998154$$

The metrics for the random activation of SAEs are summarized in Table 5.

Table 5: Summary of evaluation metrics with their respective formulas and calculated values. These metrics highlight the model's performance in terms of accuracy, precision, recall, specificity, and F1 score.

| Metric | Formula and Value |
|---|---|
| Accuracy | $\text{Accuracy} = \frac{\text{TP+TN}}{N} = 0.5$ |
| Precision | $\text{Precision} = \frac{\text{TP}}{\text{TP+FP}} = 0.5$ |
| Recall (Sensitivity) | $\text{Recall} = \frac{\text{TP}}{\text{TP+FN}} \approx 0.0000000319$ |
| Specificity | $\text{Specificity} = \frac{\text{TN}}{\text{TN+FP}} \approx 0.99999999668$ |
| F1 Score | $\text{F1} = \frac{2 \times \text{Precision} \times \text{Recall}}{\text{Precision+Recall}} \approx 0.0000000638$ |

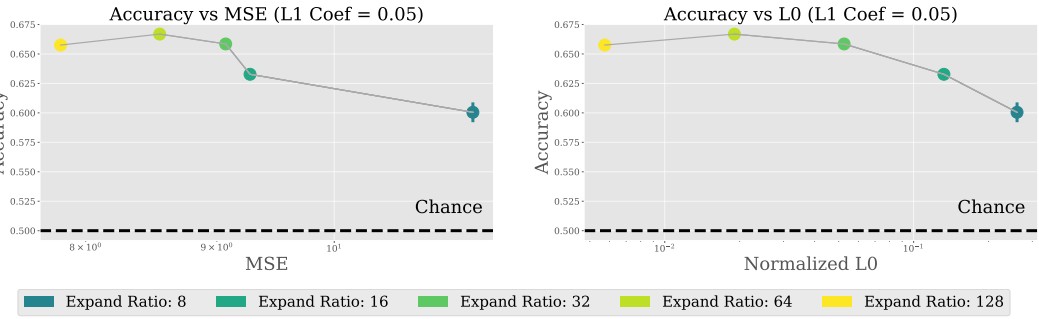

Figure 9: Comparison of Accuracy vs MSE and $L_0$ for different Expand Ratios ($L_1$ Coef = 0.05). The left panel shows the relationship between accuracy and mean squared error (MSE), while the right panel presents accuracy versus normalized $L_0$ sparsity. The markers represent various expand ratios (8, 16, 32, 64, and 128). Higher expand ratios generally show a trend towards better performance, as indicated by the yellow and green markers corresponding to expand ratios of 64 and 128. The horizontal dashed line represents the chance rate, providing a baseline for comparison.

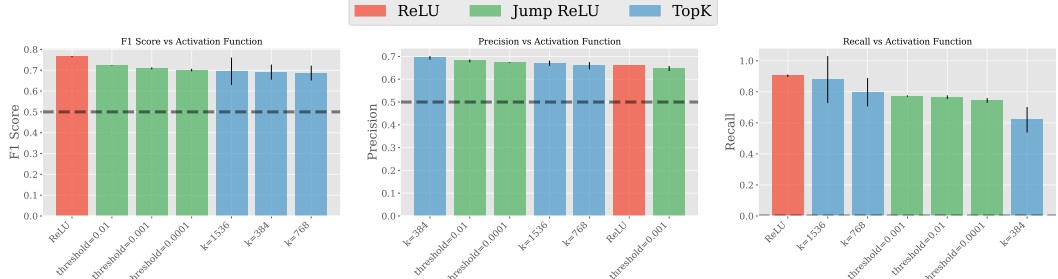

Figure 10: Comparison of F1 score, Precision, and Recall across different activation functions. The models were trained with JumpReLU (with varying thresholds), TopK (with different $k$), and standard ReLU. The results imply that ReLU may be still better than JumpReLU or TopK with the polysemous words. The error bars indicate the variation across different 3 seeds. Results for smaller $k$ of TopK and the outcomes of JumpReLU using STE are provided in Appendix D. The horizontal dashed line represents the chance rate, providing a baseline for comparison.

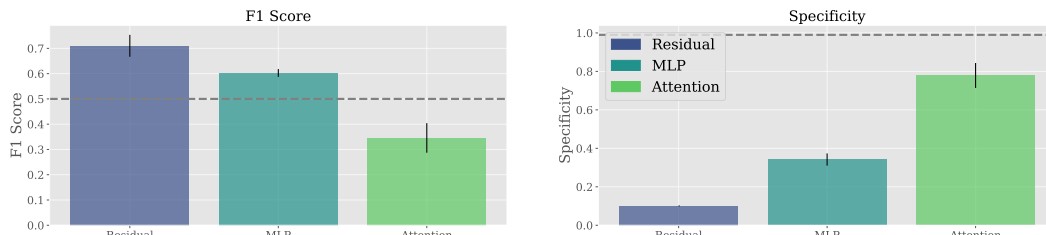

Figure 11: Comparison of F1 score and Specificity for SAEs trained on the outputs of the Residual, MLP, and Attention components. The left panel shows the F1 score, where Residual and MLP achieve higher values, while the Attention output performs lower. In contrast, the right panel shows Specificity, where the Attention output exhibits a significantly higher Specificity, despite the lower F1 score (see Appendix I for details on these components). The horizontal dashed line represents the chance rate, providing a baseline for comparison.

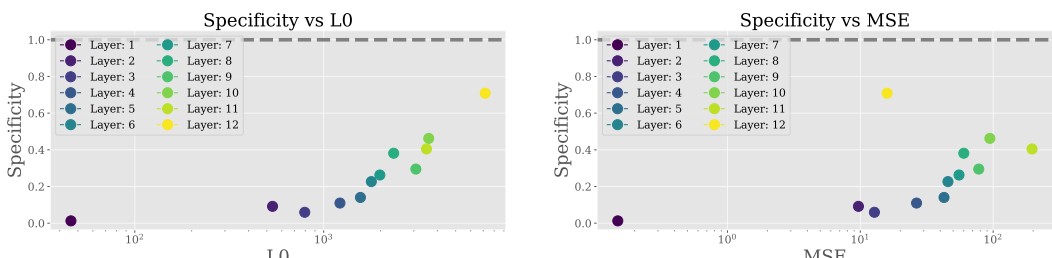

Figure 12: Comparison of Specificity across different layers of the SAE in relation to $L_0$ (left) and MSE (right). Each point represents a layer, from layer 1 to layer 12, with deeper layers (i.e., layer 6 onwards) showing an increasing trend in Specificity. This indicates that deeper layers exhibit more distinct activations for polysemous words, consistent with previous findings that deeper layers in SAEs face greater challenges in training, leading to higher MSE and $L_0$ scores. The horizontal dashed line represents the chance rate, providing a baseline for comparison

# E    DENSE SAE

To evaluate the effectiveness of SAE, we trained a comparison model, a *dense* sparse autoencoder (Dense SAE) without sparse constraints ($\lambda = 0$). The results show that the SAE consistently outperforms the Dense SAE across Accuracy, F1 score, Precision, and Recall, with a particularly significant improvement observed in Recall.

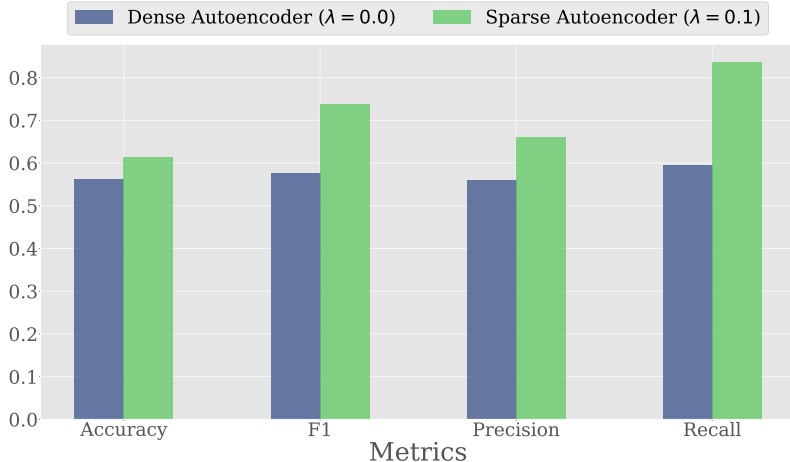

Figure 13: Comparison of evaluation metrics between dense autoencoder ($\lambda = 0$ and sparse autoencoder ($\lambda = 0.1$). Metrics include Accuracy, F1 score, Precision, and Recall. SAE consistently outperforms dense SAE across all metrics, with a particularly significant improvement in Recall.

# F    DATASET DETAILS

We show the some sample of PS-eval in Table 6. Each pair of sentences represents different meanings of the target word (e.g., "space" "save," "ball") or the same meaning. The label indicates whether the two contexts share the same meaning (Label: 1 for "Same") or differ (Label: 0 for "Different"). Poly-contexts demonstrate different meanings of the target word, while Mono-contexts show consistent meanings across sentences.

Table 6: Examples in PS-Eval.

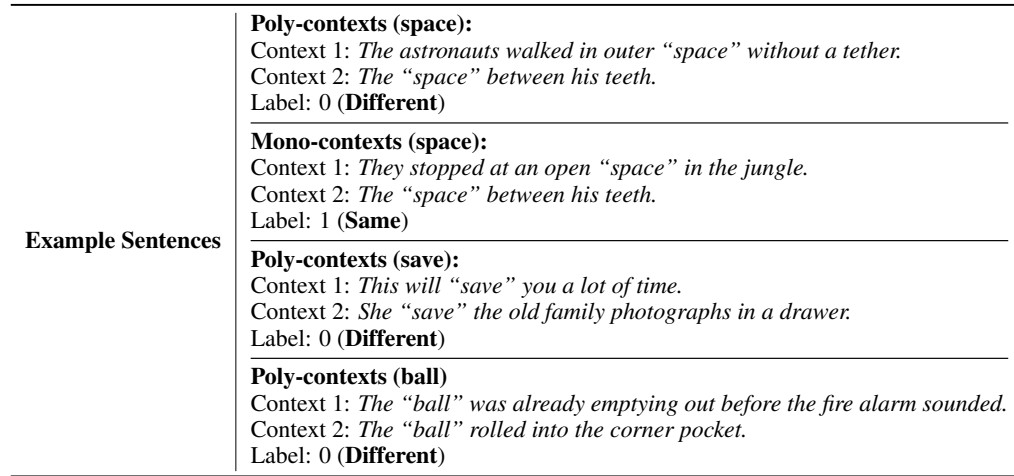

# G F1 SCORE, PRECISION, RECALL

Figure 14 shows th performance evaluation of Sparse Autoencoders (SAEs) with varying expand ratios (8, 16, 32, 64, 128) across different metrics. The top row shows F1, Precision, and Recall plotted against Mean Squared Error (MSE), while the bottom row shows F1, Precision, and Recall plotted against $L_0$ sparsity. A lower MSE or $L_0$ indicates better performance, as seen with the trend of improved F1 scores and other metrics at specific expand ratios.

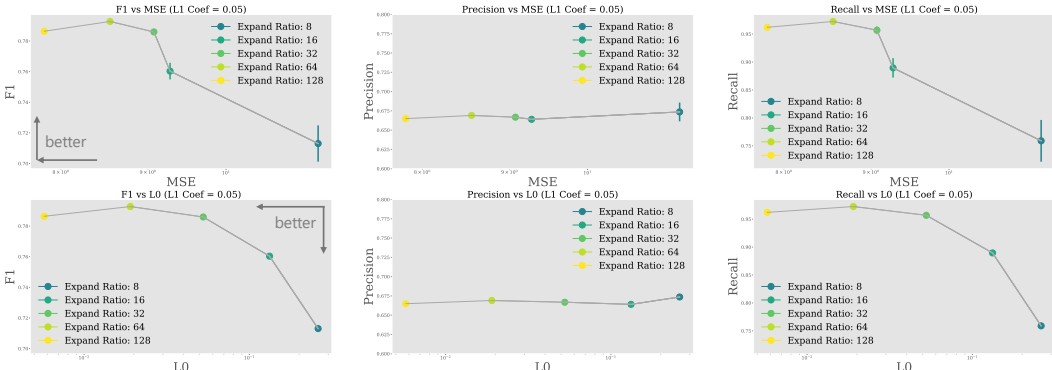

Figure 14: Comparison of F1 score, Precision, and Recall across different activation functions. The models were trained with JumpReLU (with varying thresholds), TopK (with different $k$), and standard ReLU. The results show that JumpReLU outperforms both TopK and standard ReLU in terms of F1 score, Precision, and Recall, indicating that it is better suited for capturing polysemantic representations.

## H ACTIVATION FUNCTIONS

### H.1 TOPK

The TopK activation function is designed to directly control sparsity in autoencoders by selecting only the $k$ largest activations from the latent space, while zeroing out the rest. This approach eliminates the need for the commonly used $L_1$ penalty, which approximates sparsity by shrinking activations toward zero, often leading to suboptimal representations. The TopK function is defined as:

$$z = \text{TopK}(W_{\text{enc}}(x - b_{\text{pre}}))$$

where $W_{\text{enc}}$ represents the encoder weights, $x$ is the input vector, and $b_{\text{pre}}$ is a bias term. By preserving only the top $k$ largest values, TopK maintains a fixed level of sparsity, allowing for simpler model comparisons and more interpretable representations.

### H.2 JUMPRELU

JumpReLU (Erichson et al., 2019) is an activation function introduced to enhance the performance of Sparse Autoencoders (SAEs) by improving the trade-off between sparsity and reconstruction fidelity. While the conventional ReLU activation function zeros out inputs less than or equal to zero, JumpReLU introduces a positive threshold $\theta$. Specifically, JumpReLU zeros out pre-activations below this threshold, while retaining values above it. This mechanism helps in reducing the number of false positives, thus improving sparsity, while preserving the important feature activations with high fidelity. The function is defined as:

$$\text{JumpReLU}_\theta(z) = z \cdot H(z - \theta)$$

where $H$ is the Heaviside step function and $\theta$ is a learnable threshold parameter. As illustrated in Figure 15, the JumpReLU function (blue) introduces a threshold $\theta = 1$, and only retains activations greater than this threshold, in contrast to ReLU (green), which activates any positive input.

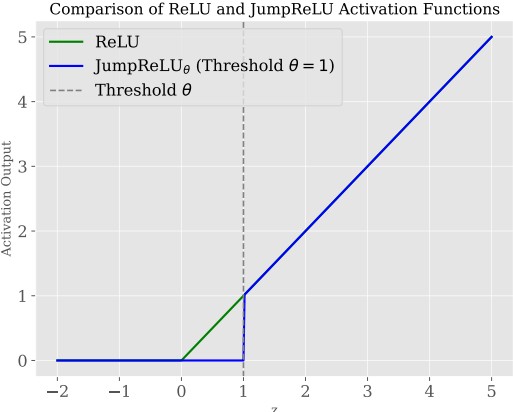

Figure 15: Comparison of ReLU and JumpReLU activation functions. The ReLU function (green) activates inputs greater than zero, while the JumpReLU function (blue) introduces a threshold $\theta = 1$, below which all inputs are zeroed out. The dashed gray line represents the threshold, illustrating how JumpReLU suppresses small activations, thereby promoting sparsity in activations.

## I    COMPONENTS OF THE TRANSFORMER AND ACTIVATION

Figure 20 illustrates the detailed experimental setup corresponding to Figure 6. The evaluation of the residual in Figure 6 is based on the SAE trained using the activations from the 'Residual output' shown in Figure 20. Similarly, the results for the MLP in Figure 6 are derived from the SAE trained on the activations from the 'MLP output' in Figure 20, and the results for the attention in Figure 20 are based on the SAE trained using the activations from the 'Attention output' in Figure 20.

This demonstrates that even within the same layer, the features captured by the SAE vary depending on the specific component (e.g., residual, MLP, or attention) whose activations are used for training.

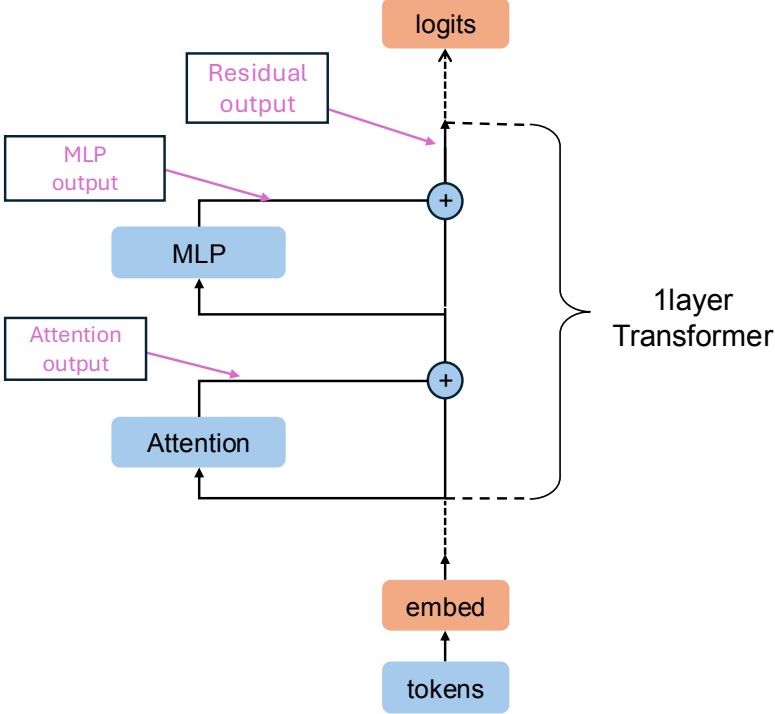

Figure 16: Illustration of a single-layer Transformer architecture. The input tokens are embedded and passed through the attention mechanism and MLP (multi-layer perceptron), with residual connections at each stage. The final output logits are computed by combining the results from the residual pathways

## J    TOPK WITH SMALLER k AND STRAIGHT THROUGH ESTIMATOR VARIANT JUMPRELU

We conducted further experiments to evaluate the impact of sparsity and activation function design on SAE performance, as detailed below:

**Results for TopK with Smaller $k$**    We extended our analysis to include TopK results for $k = d_{\text{model}}/4 = 192$ and $k = d_{\text{model}}/8 = 92$. Consistent with our expectations, smaller $k$ values achieved higher recall compared to $k = d_{\text{model}}/2 = 384$. This result further emphasizes the benefits of increased sparsity in certain evaluation contexts and highlights the importance of selecting appropriate $k$ values for optimizing recall in sparse autoencoders.

**Results for JumpReLU with STE**    We also evaluated the performance of JumpReLU using the straight-through estimator (STE) approach, following Rajamanoharan et al. (2024b). The results indicate that, even with the STE version, the improvements in metrics such as PS-eval were not as significant. This suggests that while JumpReLU offers clear advantages in terms of reconstruction and sparsity, further refinements to evaluation metrics like PS-eval may be required to fully optimize the design of sparse autoencoders.

These findings underscore the importance of carefully selecting evaluation metrics and activation functions tailored to specific goals in sparse autoencoder development. In other words, they suggest that focusing solely on achieving Pareto optimality between MSE and sparse during SAE development may overlook other critical aspects, such as interpretability.

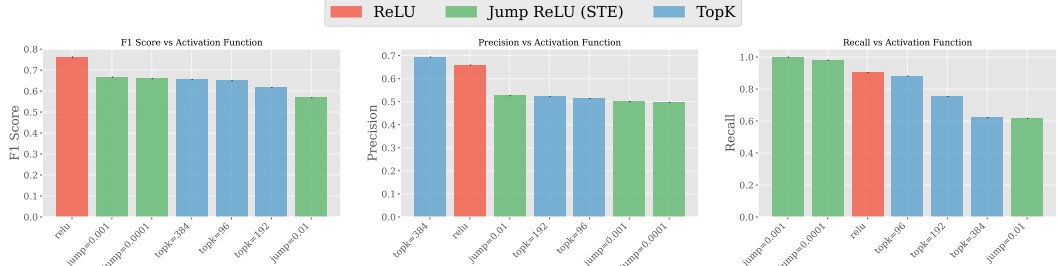

Figure 17: Comparison of activation functions (ReLU, JumpReLU (STE), and TopK) across different evaluation metrics: F1 Score, Precision, and Recall. The performance of each activation function is visualized, highlighting variations in metric outcomes based on the activation type. ReLU and JumpReLU show competitive performance, with notable differences depending on the metric.

## K    CAN THE OPEN SAE MODEL ACQUIRE MONOSEMOUS FEATURES?

Unlike some previous evaluations for SAEs that rely on the IOI task (Wang et al., 2022; Makelov et al., 2024), our PS-Eval is a model-independent and general suite for evaluation, which allows us to assess open SAEs trained on activations from models such as Pythia (Biderman et al., 2023) and Gemma (Gemma Team et al., 2024). Table 7 presents the evaluation results for SAEs trained on GPT-2 small (4 layers), Pythia 70M (4 layers), and Gemma2-2B (20 layers). While GPT-2 small closely matches the highest-performing model evaluated in our study (with an expand ratio of 64 and the ReLU activation function), other large language models (LLMs) generally received lower evaluation scores. To effectively acquire monosemantic features, our findings suggest that SAE parameters vary across different models and need to be individually tuned for each specific architecture.

Table 7: Evaluation metrics for open SAE models.

| Metric | GPT-2 small[3] | Pythia 70M [4] | Gemma2-2B [5] |
|---|---|---|---|
| Accuracy | 0.66 | 0.51 | 0.52 |
| Recall | 0.94 | 0.72 | 0.74 |
| Precision | 0.60 | 0.51 | 0.51 |
| F1 Score | 0.73 | 0.59 | 0.60 |
| Specificity | 0.38 | 0.32 | 0.30 |

## L    RELATIONSHIP BETWEEN $L_0$ REGULARIZATION AND ACTIVATION FUNCTIONS

In our setup, with an expand ratio of 32 and a model dimensionality of 768, the total number of SAE features amounts to 24,576. Experiments with the TopK activation function demonstrated that reducing the value of $k$ improves sparsity, as smaller $k$ values lead to fewer active dimensions while maintaining comparable performance metrics.

Furthermore, we observed that using JumpReLU with a jump value of 0.0001 results in the largest $L_0$-norm among all activation functions tested in this configuration. This activation function is the only one in our experiments where the number of active dimensions exceeds the original model dimensionality of 768.

Table 8: The relationship between $L_0$ regularization and various activation functions. The table compares the number of nonzero latent variables ($L_0$) for different activation functions, including ReLU, JumpReLU with two jump values (0.0001 and 0.001), and TopK with varying $k$ values $k \in \{768, 384, 192\}$. The results indicate that the choice of activation function and parameters significantly influences the sparsity of the model, as reflected in the $L_0$ values.

| | ReLU | JumpReLU-STE (jump=0.0001) | JumpReLU-STE (jump=0.001) | JumpReLU (jump=0.0001) | JumpReLU (jump=0.001) | TopK (k=768) | TopK (k=384) | TopK (k=192) |
|---|---|---|---|---|---|---|---|---|
| $L_0$ | 531 | 831 | 729 | 829 | 739 | 760 | 383 | 190 |

---

[3] https://huggingface.co/jbloom/GPT2-Small-SAEs-Reformatted
[4] https://huggingface.co/ctigges/pythia-70m-deduped__att-sm_processed
[5] https://huggingface.co/google/gemma-scope-2b-pt-res

# M  DETAILED EVALUATION METRICS

In this section, we provide additional evaluation metrics to complement the results presented in the main text. First, we enumerate possible evaluation metrics derived from the confusion matrix (see Table 9). Following the experiments in the main text, we investigate the impact of the expand ratio, the effect of layer positions, and the influence of different Transformer components on the results. The results show that the metrics that achieve higher values vary depending on the experimental variables. This highlights the need to assess the characteristics of the SAE using a variety of evaluation metrics, rather than optimizing for a single metric.

Table 9: All the possible metrics we may compute, and their interpretations.

| Metric | Formula | Interpretation |
|--------|---------|----------------|
| Accuracy | $\frac{TP+TN}{TP+FP+TN+FN}$ | How well the model predicts the overall situation |
| Recall | $\frac{TP}{TP+FN}$ | When the same meaning's words are given, how often they are identified as the same |
| Precision | $\frac{TP}{TP+FP}$ | When the same features are activated, how often the meanings are same |
| Specificity | $\frac{TN}{TN+FP}$ | When the different meaning's words are given, how often they are identified as different |
| Sensitivity | $\frac{TN}{TN+FN}$ | When the different features are activated, how often the meanings are different |
| S-F1 | $\frac{2 \times Recall \times Precision}{Recall+Precision}$ | How Recall and Precision are balanced |
| D-F1 | $\frac{2 \times Specificity \times Sensitivity}{Specificity+Sensitivity}$ | How Specificity and Sensitivity are balanced |
| Average-F1 | $\frac{1}{2}\left(\text{S-F1} + \text{D-F1}\right)$ | How S-F1 and D-F1 are balanced |

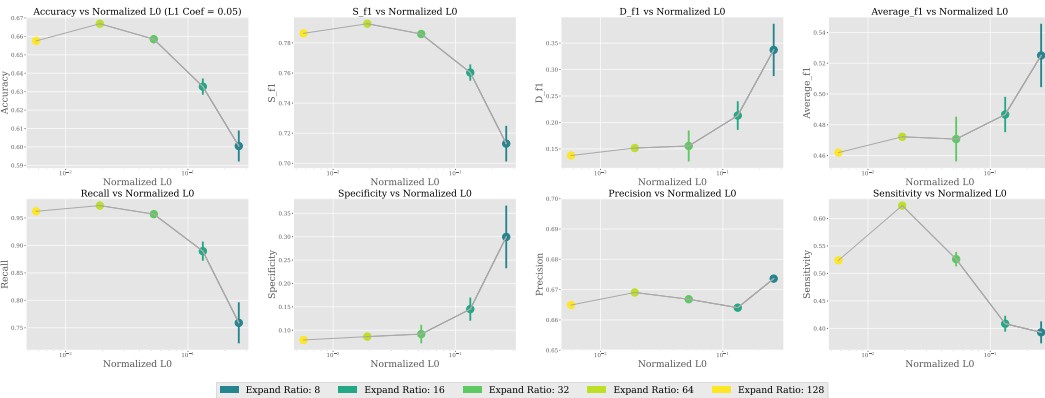

Figure 18: Evaluation of various performance metrics (Accuracy, Recall, Specificity, Precision, Sensitivity, and F1 scores) as a function of normalized $L_0$ sparsity across different expand ratios (8, 16, 32, 64, 128).

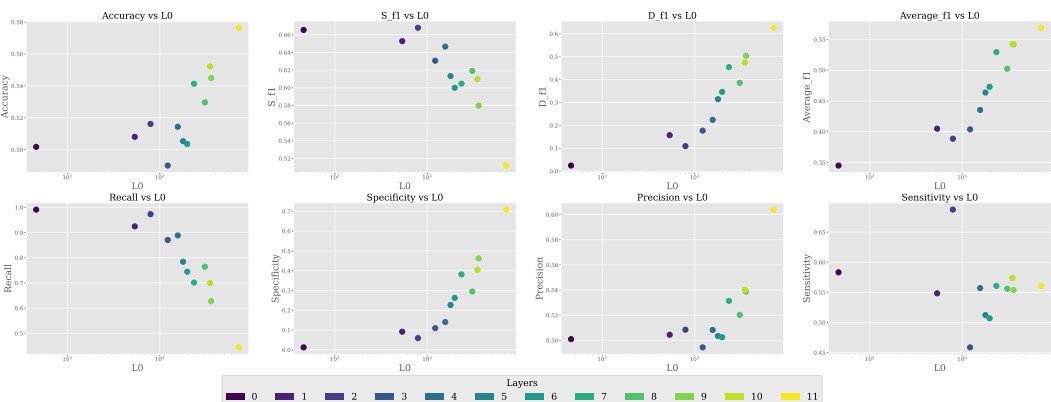

Figure 19: Evaluation of various performance metrics (Accuracy, Recall, Specificity, Precision, Sensitivity, and F1 scores) as a function of normalized $L_0$ sparsity across different Layers.

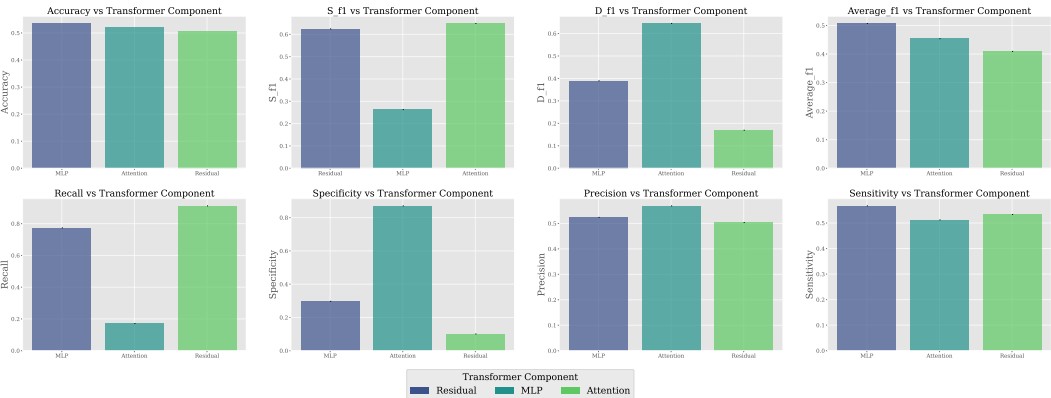

Figure 20: Evaluation of various performance metrics (Accuracy, Recall, Specificity, Precision, Sensitivity, and F1 scores) as a function of normalized $L_0$ sparsity across different transformer component.

# N INTERPRETATION AND EVALUATION OF LOGIT LENS RESULTS WITH LLMS

We are exploring ways to extend this qualitative analysis into a more quantitative evaluation. One potential approach is leveraging automated evaluation methods using LLMs, as demonstrated in works like Cunningham et al. (2023). We conducted a preliminary experiment using the results from Table 3 of our paper. Specifically, we focused on the example involving the word "space". In this case, **Context 1** refers to "space" in the sense of "universe" or "outer space," while **Context 2** refers to "space" as in "blank space" or "gap." The input provided to GPT-4 was as follows:

```
## Instruction:
You will be provided with two contexts containing the same target word.
Additionally, related words for the target word will also be given.

1. Determine whether the target word has the same meaning
    or different meanings in these two contexts.

2. Explain the meaning of the target word in each context.

## Output Format:
- Same or Different: [Your answer]
- Meaning in Context 1: [Your explanation]
- Meaning in Context 2: [Your explanation]

## Context1 related words
flight, plane, shuttle, gravity, craft, Engineers, planes

## Context2 related words
Layout, occupied, spaces, vacated, space, shuttle, occupancy
```

The following output was generated by GPT-4:

```
Output:
Same or Different: Different

Meaning in Context 1: The target word refers to a vehicle or object
designed for travel or transportation in outer space,
such as a spacecraft or shuttle.

Meaning in Context 2: The target word refers to a physical area or
expanse that can be occupied orleft empty, often used
in the context of spatial arrangements or occupancy.
```

The results suggest that GPT-4 can interpret the logit lens outputs from the SAE features in a manner consistent with human intuition. Specifically, it can distinguish whether a feature corresponds to the same or different meanings. Additionally, its reasoning aligns with plausible interpretations, such as identifying that **Context 1** refers to "universe" or "outer space," while **Context 2** refers to "blank space."

Using an LLM (GPT-4) as an evaluator opens new possibilities for extending the implications of the logit lens results. By systematically applying such methods, we can quantitatively validate the alignment of SAE features with human interpretations, addressing this question in a rigorous and scalable manner.

## O    COMPARISON OF SAES TRAINED WITH DIFFERENT LAYERS

In this section, we present results comparing different layers within the network. Prior studies often select a particular layer somewhat arbitrarily. To verify that the trends and conclusions we draw are not restricted to a specific layer, we conduct additional experiments using layer 6 as an alternative input feature source, as described in Section 6.

### O.1    EXPERIMENTS USING LAYER 6

Here, we replicate the experiments with the proposed method by substituting layer 6 for layer 4. The accuracy results for different expand ratios are shown in Table 10.

As the expand ratio increases, we observe consistent improvements in accuracy. Importantly, the overall patterns remain broadly similar regardless of whether we use layer 4 or layer 6. This suggests that the choice of layer does not substantially influence the observed trends, reinforcing the robustness of our findings.

Table 10: Relationship between Expand Ratio and Accuracy for Layer 4 and Layer 6 (Figure 3)

| Expand Ratio | 8 | 16 | 32 | 64 | 128 |
|---|---|---|---|---|---|
| Layer 4 | 0.6005 | 0.6328 | 0.6585 | 0.6669 | 0.6576 |
| Layer 6 | 0.4982 | 0.6203 | 0.6370 | 0.6550 | 0.6686 |

### O.2    COMPARISON OF ACTIVATION FUNCTIONS ACROSS LAYERS

Next, we compare the impact of different activation functions across layers. In addition to ReLU, we consider STE with varying jump parameters and Top-$k$ methods. The F1 scores, precision, and recall for these configurations are shown in Tables 11, 12, and 13, respectively.

The results consistently show that ReLU achieves the highest F1 score, regardless of which layer is chosen. This outcome aligns with the findings obtained from layer 4 alone. Moreover, the trends in performance across different activation functions do not substantially change when using layer 6 instead of layer 4, indicating that our conclusions hold broadly across layers.

Table 11: F1 Scores for Different Activation Functions (Figure 5)

| Method | ReLU | STE(jump=0.001) | STE(jump=0.0001) | topk=384 | topk=96 | topk=192 |
|---|---|---|---|---|---|---|
| Layer 4 | 0.7623 | 0.6667 | 0.6590 | 0.6550 | 0.6490 | 0.6166 |
| Layer 6 | 0.7509 | 0.6203 | 0.6136 | 0.5985 | 0.6203 | 0.6316 |

Table 12: Precision for Different Activation Functions (Figure 5)

| Method | topk=384 | ReLU | topk=192 | topk=96 | STE(jump=0.001) | STE(jump=0.0001) |
|---|---|---|---|---|---|---|
| Layer 4 | 0.6939 | 0.6593 | 0.5218 | 0.5251 | 0.5000 | 0.4964 |
| Layer 6 | 0.5170 | 0.6729 | 0.5178 | 0.5136 | 0.4901 | 0.4599 |

Table 13: Recall for Different Activation Functions (Figure 5)

| Method | STE(jump=0.001) | STE(jump=0.0001) | ReLU | topk=96 | topk=192 | topk=384 |
|--------|-----------------|------------------|------|---------|----------|----------|
| Layer 4 | 0.9951 | 0.9802 | 0.9034 | 0.8813 | 0.7536 | 0.6203 |
| Layer 6 | 0.9501 | 0.8615 | 0.8094 | 0.8523 | 0.8094 | 0.7104 |

## P  EXTENDED DISCUSSION

**Extending PS-Eval Beyond Natural Language**  While SAE was initially proposed as a tool to analyze polysemantic activations in language models, its utility has recently expanded to other domains, such as diffusion models and vision-language models like CLIP (Surkov et al., 2024; Lee, 2024). These developments suggest opportunities to adapt and extend the PS-Eval framework to new contexts. For instance, Surkov et al. (2024) explores the application of SAE in the Stable Diffusion text-to-image model. Given the reliance of this model on text inputs, PS-Eval-like methods could be employed to analyze its responses to polysemous text prompts, providing insights into its semantic representation capabilities. Similarly, for vision-based models like CLIP (Lee, 2024), the PS-Eval framework could be extended by leveraging polysemous image inputs; i.e. images with multiple interpretations. Datasets such as the optical illusion dataset proposed in Shahgir et al. (2024) offer a promising avenue for evaluating how vision-language models handle input ambiguity, serving as a foundation for adapting PS-Eval to visual modalities. These examples highlight the versatility of the PS-Eval framework and its potential applicability beyond natural language processing, provided that input ambiguity and context-specific challenges are carefully addressed in each domain.

**Evaluation of Open SAEs**  We extensively evaluate SAEs from GPT-2-small and also assess the quality of open SAEs from GPT-2-small, Pythia 70M, and Gemma2-2B in Appendix K. The results imply that open SAE trained from larger/more recent LLMs does not always achieve better performance on PS-Eval (such as GPT-2-small > Gemma2-2B). Bereska & Gavves (2024) mention that whether larger and more recent LLMs produce activation patterns that are more interpretable from the perspective of SAE remains unclear despite their better capability on natural language tasks. Capable/large-scale LLMs may have more complex activation internally. Our results in Appendix K are aligned with their statement. In principle, we should assume that the performances of LLMs and SAEs are independent. It is not obvious whether the interpretability of SAE correlates with the capability of base LLMs, and we think it is worth investigating in future works.

**Mechanistic Interpretability**  In the field of mechanistic interpretability, research spans from comprehensive *circuit* analysis in toy models, such as grokking (Power et al., 2022; Nanda et al., 2023; Furuta et al., 2024; Minegishi et al., 2024), to methods like SAE for analyzing *features* in large-scale models such as LLMs (Wang et al., 2022; Lieberum et al., 2023; Gao et al., 2024). PS-Eval, as proposed, provides a practical evaluation framework for model-independent SAE assessments, bridging the gap between theoretical insights and practical applicability.

