# OpenReview forum: "Rethinking Evaluation of Sparse Autoencoders through the Representation of Polysemous Words"
_ICLR.cc/2025/Conference — ICLR 2025 Poster_

### Official Review · Reviewer_k8GD · 2024-10-26

**Soundness:** 2
**Presentation:** 2
**Contribution:** 3
**Rating:** 6
**Confidence:** 5

**Summary:**

In the literature, SAEs are often evaluated on MSE and L0. The paper introduces a new metric for evaluation of SAEs that directly measures whether SAEs learn separate features for the different meanings of polysemous words (words which have different meanings based on the context). The metric is constructed from the WiC dataset (filtering for words which tokenize to single tokens). The paper finds that improving the MSE-L0 frontier does not necessarily lead to improvements on this newly introduced metric. The paper also shows that eventually, scaling SAEs no longer improves score on this metric, that alternative activation functions are not always better, and that later layers

**Strengths:**

- Well motivated metric. Correct in identifying problems with MSE/L0 as a metric of SAE quality. Using polysemy as a source of eval is a good idea.
- Confusion matrix approach is reasonable and consideration of metrics like F1 provides more granularity.

**Weaknesses:**

- k=384 is an extremely high k for TopK. Gao et al. (2024) recommends using d_model / 2 for the _maxk_ hyperparameter, not themain k hyperparameter. Ideally, a much smaller k is used for $k$. Using a value of k equal to or larger than d_model doesn't make much sense.
- JumpReLU comparison would be stronger if the STE variant from Rajamanoharan et al. (2024) were used.

**Questions:**

- Normalized L0 (L0 / N) is less informative as a metric than L0 because the amount of information in the latents is roughly proportional to L0 * log N, so across different N's, L0 is roughly comparable, whereas L0 / N is not very comparable.
- How are the SAEs initialized? Prior work (Conerly et al., 2024; Gao et al., 2024) has found that initializing the encoder to the transpose of the decoder is very important for scaling SAEs.
- What are the L0 values attained by the SAEs for each activation function? Are they in a similar range? Are they less than d_model?
- How many dead latents are there in the SAEs?

---

> ### Author Response · Authors · 2024-11-21
> **Author Response (1/2)**
>
> We thank the reviewer for your constructive feedback and suggestions. Please let us know if our responses in the following address your concerns.
>
> We revised the paper based on the reviewers’ comments, and the major edit was highlighted with coloring (purple). Please also check the updated manuscript.
>
>
> **W1 & W2**
>
> > Ideally, a much smaller k is used for k. Using a value of k equal to or larger than d_model doesn't make much sense.
>
> > JumpReLU comparison would be stronger if the STE variant from Rajamanoharan et al. (2024) were used.
>
> As per your suggestions, we have included the following comparison in **Appendix K**:
>
> 1. **Results for TopK with smaller $ k $:**
>    - Specifically, $ k = d_{\mathrm{model}}/4 = 192 $ and $ k = d_{\mathrm{model}}/8 = 92 $.
>    - As you correctly pointed out, smaller $ k $ values achieve higher recall compared to $ k = d_{\text{model}}/2 = 384 $, further emphasizing the benefits of increased sparsity in certain evaluation contexts.
>
> 2. **Results for jumpReLU using STE:**
>    - Even with the STE version of jumpReLU, the improvements in the evaluated metrics (e.g., PS-eval) were not as significant.
>    - This suggests that while jumpReLU offers benefits in reconstruction and sparsity, other metrics like PS-eval might require further refinement to better develop an ideal SAE.
>
> These findings underscore the importance of carefully selecting evaluation metrics and activation functions to optimize for specific goals in sparse autoencoder development.
> Please let us know if there are additional aspects you’d like us to expand upon.
>
>
> **Q1**
>
> > Normalized L0 (L0 / N) is less informative as a metric than L0 because the amount of information in the latents is roughly proportional to L0 * log N, so across different N's, L0 is roughly comparable, whereas L0 / N is not very comparable.
>
> Thank you for your insightful feedback. We agree that normalized L0 (L0 / N) may be less informative in this context. Based on your suggestion, we updated Figures 3 and 5 to use non-normalized L0 on the x-axis.
>
> We found that the results remain consistent with the original trends. Specifically, Figure 3 shows that increasing the expand ratio leads to better outcomes. This adjustment provides a clearer and more interpretable representation of the data.
>
>
>
> **Q2**
>
> > How are the SAEs initialized? Prior work (Conerly et al., 2024; Gao et al., 2024) has found that initializing the encoder to the transpose of the decoder is very important for scaling SAEs.
>
> We initialize the weights of the Decoder as the transposed weights of the Encoder, following the methodology from previous studies.
> To clarify, we have added a detailed explanation of this procedure in **Appendix B.**

---

> > ### Author Response · Authors · 2024-11-21
> > **Author Response (2/2)**
> >
> > **Q3**
> >
> > > What are the L0 values attained by the SAEs for each activation function? Are they in a similar range? Are they less than d_model?
> >
> > In the revision,  we examined the $ \mathrm{L}_0 $-norm across different activation functions.
> > The detailed were added to **Appendix L.**
> >
> > In our setup, with an expand ratio of 32 and a model dimensionality of 768, the total number of SAE features is 24,576. As you suggested in your comments under Weakness, reducing the value of k in the TopK activation function indeed improves sparsity.
> >
> > Additionally, we observed that when using JumpReLU with a jump value of 0.0001, the resulting
> > L0-norm is the largest among all activation functions tested in this setup. Notably, this activation function is the only one in our experiments where the number of active dimensions exceeds the original model dimensionality of 768. This unique behavior highlights the potential of JumpReLU for promoting sparsity while still achieving significant activation.
> >
> >
> > | Activation Function  |  ReLU | JumpReLU (jump=0.0001) | JumpReLU (jump=0.001) | TopK (k=768) | TopK (k=384) | TopK (k=192) |
> > |:--------------------:|:--------:|:-----------------:|:----------------:|:----------:|:----------:|:----------:|
> > | L0                  |   531    |       829         |       739        |     760    |     383    |     190    |
> >
> >
> >
> >
> > **Q4**
> >
> > > How many dead latents are there in the SAEs?
> >
> > In our study, we have incorporated an auxiliary loss called Ghost grads by default, following prior research (Gao et al., 2024). To address your query, we have added an analysis of the number of dead latents during training to **Appendix A**. Without Ghost Grads, nearly 10,000 dead latents arise out of the 24,576 (32*768) features in the SAE. However, with Ghost Grads, the number of dead latents is effectively reduced to almost zero.
> > Furthermore, the evaluation results using PS-eval also showed better performance when Ghost Grads were applied, reinforcing our decision to include Ghost Grads as the default setting in our study.
> >
> > Thank you again for your thoughtful comments. We hope our responses address your concerns, and we are happy to provide further clarification if needed.

---

> > > ### Author Response · Authors · 2024-11-26
> > > **A Reminder to Reviewer k8GD**
> > >
> > > Again, thank you for your valuable review and constructive feedback on our paper. As we have not yet received a response, we would like to kindly remind Reviewer k8GD of our efforts to address your comments. To answer your questions, we have added new experiments to the appendix and updated the figures in the main text to improve clarity.
> > >
> > > We would sincerely appreciate it if you could kindly assess whether our revisions have effectively addressed your concerns.

---

> > > ### Comment · Reviewer_k8GD · 2024-11-26
> > >
> > > - The results for smaller k's are appreciated, strengthening the result. I'm confused about the JumpReLU-STE results - what does the jump hyperparameter indicate in that case? Unlike in normal JumpReLU, the STE variant does not have a hyperparameter for the location of the jump, but rather the location of the jump is learned and the sparsity is controlled with a L0 coeffivient hyperparameter.
> > > - The use of L0 instead of normalized L0 and clarification about initialization and dead latents is appreciated.
> > > - It would be much better presentation-wise for the results in appendix K to be combined with figure 4, and for appendix L to list sparsities for all activation functions considered (e.g including JumpReLU-STE). With this change, I would increase the presentation score to 3.
> > > - However, the L0 values appear to vary quite substantially across the different methods, which makes the comparison between different methods somewhat harder to interpret. For example, it's unclear whether ReLU performs better on F1 than JumpReLU because ReLU is better than JumpReLU, or if it's confounded by the sparsity. The additional topk results improve soundness on this front, but it would further strengthen the results to have more overlap between the sparsity ranges of the different methods (e.g results with JumpReLU with larger threshold, ReLU with L1 coefficient swept) -- especially for the L0 < d_model regime (the L0 > d_model regime is quite unusual, and difficult to interpret). I think this should be listed as a limitation, or ideally, experiments with more ReLU L1 values and JumpReLU thresholds are conducted to ensure more overlap in L0 ranges between different activations.

---

> > > > ### Author Response · Authors · 2024-11-27
> > > > **Author Reply**
> > > >
> > > > We thank the reviewer for your response. We answer your follow-up questions as follows.
> > > >
> > > > > The results for smaller k's are appreciated, strengthening the result. I'm confused about the JumpReLU-STE results - what does the jump hyperparameter indicate in that case? Unlike in normal JumpReLU, the STE variant does not have a hyperparameter for the location of the jump, but rather the location of the jump is learned and the sparsity is controlled with a L0 coeffivient hyperparameter.
> > > >
> > > > We apologize for the slightly confusing explanation.
> > > >
> > > > In the STE variant, our understanding is that the threshold `jump` is also optimized (Rajamanoharan et al., [1], Section 4).
> > > >
> > > > In JumpReLU-STE results, the `jump` represents the initial value of this learnable threshold. The implementation was based on the pseudo-code provided in Appendix J of Rajamanoharan et al.,[1].
> > > >
> > > > We have explicitly mentioned this point in Section 6.3 for clarity.
> > > >
> > > > [1] Rajamanoharan et al., Jumping Ahead: Improving Reconstruction Fidelity with JumpReLU Sparse Autoencoders. 2024. https://arxiv.org/abs/2407.14435
> > > >
> > > > ----
> > > > > The use of L0 instead of normalized L0 and clarification about initialization and dead latents is appreciated
> > > >
> > > > We have expanded Section 6.2 to provide additional details about the use of L0.
> > > > In addition, we included a description of initialization and dead latents in Section 3.
> > > >
> > > > ----
> > > > > It would be much better presentation-wise for the results in appendix K to be combined with figure 4, and for appendix L to list sparsities for all activation functions considered (e.g including JumpReLU-STE). With this change, I would increase the presentation score to 3.
> > > >
> > > > We thank the reviewer for your suggestion. We have incorporated the results from Appendix K, including those for STE JumpReLU and smaller `k` values for TopK, into Figure 4 to improve the presentation.  Additionally, we have updated Appendix L to include the L0 results for each activation function, including JumpReLU-STE.
> > > >
> > > > The table of L0 results is provided below.
> > > >
> > > > | Activation Function      | ReLU|JumpReLU-STE (jump=0.0001) | JumpReLU-STE (jump=0.001) | JumpReLU (jump=0.0001) | JumpReLU (jump=0.001) | TopK (k=768) | TopK (k=384) | TopK (k=192) |
> > > > |--------------------------|---------------------------|---------------------------|------------------------|----|-------------------|--------------|--------------|--------------|
> > > > | **L0**                  | 531                       | 831                       | 729                    | 829          |739         | 760          | 383          | 190          |
> > > >
> > > >
> > > > -----
> > > > > I think this should be listed as a limitation, or ideally, experiments with more ReLU L1 values and JumpReLU thresholds are conducted to ensure more overlap in L0 ranges between different activations.
> > > >
> > > >
> > > > Currently, the comparison between activation functions does not fully disentangle the effects of sparsity from the effects of the activation functions themselves. Due to time constraints, we are unable to include additional results in the main PDF at this time. However, we plan to conduct experiments with more ReLU L1 coefficients and JumpReLU thresholds to achieve better overlap in L0 ranges between different activations (especially in the L0 < d_model regime).
> > > >
> > > > We will follow up with these results in a markdown format within the next few days and ensure they are included in the final version of the paper.

---

> > > > > ### Author Response · Authors · 2024-12-02
> > > > >
> > > > > > I think this should be listed as a limitation, or ideally, experiments with more ReLU L1 values and JumpReLU thresholds are conducted to ensure more overlap in L0 ranges between different activations.
> > > > >
> > > > > To address the concern about disentangling the effects of sparsity and activation functions, we have conducted additional experiments with various $ L_1 $ coefficients for ReLU and different thresholds for JumpReLU. These experiments reveal that ReLU consistently outperforms JumpReLU in terms of F1 score, regardless of the $ L_0 $ sparsity range, including cases where $ L_0 < d_{\text{model}} = 768 $.
> > > > >
> > > > > **ReLU with Swept $ L_1 $ Coefficients**
> > > > >
> > > > > We performed a sweep of $ L_1 $ coefficients to expand the range of sparsity levels for ReLU.
> > > > >
> > > > > | $ L_1 $ Coefficient | 0.005  | 0.01   | 0.05   | 0.1    | 0.5    |
> > > > > |-----------------------|--------|--------|--------|--------|--------|
> > > > > | F1 Score             | 0.6364 | 0.6437 | 0.7623 | 0.7509 | 0.6729 |
> > > > > | $ L_0 $ Norm       | 1642   | 1235   | 531    | 381    | 253     |
> > > > >
> > > > > **JumpReLU with Larger Thresholds**
> > > > >
> > > > > For JumpReLU, we increased the jump threshold to explore lower sparsity ranges ($ L_0 < d_{\text{model}} = 768 $).
> > > > >
> > > > > | Jump Threshold | 0.0001 | 0.001  | 0.01   | 0.05   | 0.1    |
> > > > > |----------------|--------|--------|--------|--------|--------|
> > > > > | F1 Score       | 0.6541 | 0.6667 | 0.6525 | 0.6320 | 0.6012 |
> > > > > | $ L_0$ Norm | 831    | 729    | 699    | 651    | 572    |
> > > > >
> > > > > We hope these additional findings address your concerns and provide further clarity on the differences between these activation functions. Thank you for your valuable feedback.

---

### Official Review · Reviewer_fi7z · 2024-11-01

**Soundness:** 2
**Presentation:** 2
**Contribution:** 2
**Rating:** 6
**Confidence:** 3

**Summary:**

This paper introduces PS-Eval, a new method for assessing whether SAEs can extract monosemantic features from transformer models by examining their handling of polysemous words. While the paper offers valuable contributions through its evaluation methods, empirical analysis across different architectures, and interesting findings about layer depth and attention mechanisms, it suffers from significant conceptual limitations. My core concern is the paper's unexamined assumption that SAE features should directly map to word meanings, along with questions about whether insights from polysemous words can generalise to broader SAE evaluation. The empirical work is relatively thorough and the findings are interesting, particularly regarding the role of attention in handling polysemy and the logit lens analysis, but the theoretical framework needs strengthening and the scope may be too narrow. I recommend rejection in its current form while strongly encouraging resubmission after addressing these conceptual issues, improving the theoretical framework, and clarifying the presentation.

**Strengths:**

__Novel evaluation methodology__:
* Clear metrics based on confusion matrix approach
* Well-defined dataset construction from WiC
* Systematic evaluation framework that could be extended to other models


__Experimental analysis__:
* Thorough comparison across different activation functions
* Valuable insights about layer depth and transformer components
* Interesting findings about attention's role in handling polysemy
* Useful logit lens analysis demonstrating semantic differentiation


__Technical contributions__:
* Model-independent evaluation metric
* Practical insights for SAE deployment
* Clear demonstration of limitations in current MSE-$L_0$ optimisation

**Weaknesses:**

__Core conceptual concerns__: The paper assumes SAE features should directly map to word meanings. This is a strong assumption that needs more justification. It's possible that effective SAEs might represent meaning through other feature combinations. The focus on polysemous words, while interesting, may be too narrow to support broad conclusions about SAE quality.


__Methodology limitations__:
* The evaluation method could benefit from more discussion of edge cases and failure modes
* Limited exploration of how results generalise beyond the specific test cases
* Need for clearer discussion of how PS-Eval relates to existing SAE objectives


__Presentation issues__:
* Some key terms need clearer definition, particularly in the introduction (e.g. "semantics-focused evaluation", "polysemous words")
* Several figures would benefit from more detailed captions
* Some sections (particularly the transformer component analysis) need more thorough explanation
* Quite a few typos (e.g. "assesse", "spolysemantic", and incomplete sentences e.g. "In our evaluation across different transformer components, specifically the residual layers, MLPs and self-attention.")

**Questions:**

1. How do you justify the assumption that SAE features should directly correspond to word meanings? Could you discuss alternative feature organisations that might be equally valid?
* For instance, it's easy to imagine an SAE feature that fires on a particular word (e.g. "tweet") regardless of context, and another SAE feature fires on sentences regarding birds and another fires on sentences regarding social media (e.g. "Twitter"), distinguishing the two. It's not clear to me that this is a strong failure mode of an SAE.
* I don't know if we should expect SAEs to learn features directly corresponding to meaning (the strong linear representation hypothesis) because we don't know what the atomic units (i.e., features) of a model's computation actually are.
2. Could you elaborate on how insights from polysemous word evaluation might generalise to other aspects of SAE quality?
3. The logit lens analysis shows promising results - could you expand on how this validates your evaluation method?

---

> ### Author Response · Authors · 2024-11-21
> **Author Response (1/4)**
>
> We thank the reviewer for careful reading and thoughtful suggestions.  Please let us know if our responses in the following address your concerns.
>
> We revised the paper based on the reviewers’ comments, and the major edit was highlighted with coloring (purple). Please also check the updated manuscript.
>
>
> **Core conceptual concerns & Q1**
>
> > The paper assumes SAE features should directly map to word meanings. This is a strong assumption that needs more justification.
>
> > How do you justify the assumption that SAE features should directly correspond to word meanings?
>
> Considering the previous literature on SAEs, we believe that it is a reasonable assumption that the significant SAE features are directly relevant to word meanings, which are well-supported by the empirical observations. We justify our assumption on the SAE features from three perspectives:  (1) findings from previous studies, (2) our experimental design, and (3) the results of our logit lens analysis.
>
> **Findings in Prior Works**
>
> It is well-established in prior research that SAE features map to word meanings.
> For instance, in Templeton et al., (2024) [1], features corresponding to specific words such as "Golden Gate Bridge" or "Brain Science" were identified.
> Similarly, GPT-4 evaluations in Table 1 of Cunningham et al., (2023) [2] demonstrated that SAE features correspond to word meanings such as "legal terms."
> More recently, Li et al., (2024) [3] analyzed the geometric structure of concepts in LLMs under the assumption that SAE features encode **semantic concepts**, ranging from specific word meanings such as "king" and "queen" to more abstract concepts like "code" and "math."
> These findings align with our underlying assumption that SAE features respond specifically to word meanings.
>
> **Our Experimental Design**
>
> As you noted in your "tweet" example, we acknowledge that, depending on the context, SAE features may fire for elements beyond word meanings. To address this, our experiments are specifically designed to embed word meanings into SAE features. As described in Section 4.2, SAE features were trained on activations of the target word within the prompt, such as:
>
> `{context}. The {target_word} means `
>
> This design ensures that the meaning of the target word is directly incorporated into the SAE features. By tailoring our framework in this way, we aimed to align SAE features closely with “word meaning”.
>
>
> **Results of the Logit Lens Analysis**
>
> Building on the experimental design above, our results further validate that SAE features correspond to word meanings. As shown in Table 3, SAE features fire based on the meanings of words. For example, in the case of the token "space" (Table 3, left column), the feature that activates depends on the context, indicating distinct meanings: "space" as "outer space" (green) and "space" as "blank space" (orange).
>
> In conclusion, the concern regarding the "strong assumption that our paper directly maps SAE features to word meanings" is addressed as follows: (1) Prior research supports the expectation that SAE features can capture word meanings, aligning with our assumptions. (2) Our experimental design was intentionally structured to encourage SAE features to reflect word meanings by tailoring the framework to this specific goal. (3) The results of logit lens confirm that the SAE features successfully capture word meanings as intended. While we agree that SAE features may also fire for aspects other than word meanings because we don't know what the actual atomic units of computation in a model are, based on these empirical observations and findings, we also believe that the assumption in question is well-justified.
>
> In response to this concern, we have restructured the manuscript to clarify the validation of our assumption. The logit lens results have been moved to Section 5, where they are now explicitly positioned as evidence supporting the assumption that SAE features map to word meanings. This reorganization highlights the role of these results in demonstrating how SAE features align with word meanings, providing a clearer narrative flow.
>
> Please refer to the revised version for the updated section structure and details. If there are any aspects that remain unclear, please do not hesitate to let us know.
>
> - [1] Templeton et al., Scaling Monosemanticity: Extracting Interpretable Features from Claude 3 Sonnet, https://transformer-circuits.pub/2024/scaling-monosemanticity/index.html
> - [2] Cunningham et al., Sparse Autoencoders Find Highly Interpretable Features in Language Models. https://arxiv.org/abs/2309.08600
> - [3] Li et al., The Geometry of Concepts: Sparse Autoencoder Feature Structure. https://arxiv.org/abs/2410.19750

---

> ### Author Response · Authors · 2024-11-21
> **Author Response (2/4)**
>
> > The focus on polysemous words, while interesting, may be too narrow to support broad conclusions about SAE quality.
>
> Our proposed PS-eval is an evaluation metric to assess whether SAEs successfully capture “word meanings”, which has been not evaluated or considered when inventing better SAEs in previous works (focusing on sparsity, compressibility, functionality). To enhance the overall quality of SAEs, we believe that a multi-faceted evaluation approach incorporating various metrics is mandatory. Rather than fully replacing the existing evaluations (e.g. $\mathrm{L}_0$, MSE, IOI performance), our paper intends to use PS-Eval as a complemental evaluation of learned SAEs to take the semantics quality into consideration. PS-eval addresses a critical gap in the evaluation of SAEs by enabling direct assessment of semantic representations.
>
> To emphasize this, we revised the related work section (**Section 2**) to clarify the scope of our study (highlighted in purple).
>
> **Methodology limitations**
>
> > The evaluation method could benefit from more discussion of edge cases and failure modes
>
> We found that some key edge cases and failure modes of PS-Eval lie in their reliance on evaluating SAE features based on maximum activation. As illustrated in Section 4.2 and Figure 1 (left), PS-Eval is designed to evaluate how well SAE features respond to LLM activations corresponding to the meanings of target words. However, it is not guaranteed that the feature with the maximum activation always corresponds to the "word meaning" in a context, and the current protocol ignores other non-maximum activations.
>
> For example, as shown in the logit lens results in Table 3, while there are cases where SAE features fire strongly for word meanings, this may not consistently occur across all samples or LLMs. In some cases, the feature with the highest activation may capture an unrelated aspect rather than the intended word meaning (such as splitted tokens).
>
> We included these points in the revised version under the Discussion and Limitation section in **Appendix M** to encourage future research directions.
>
> **Q2**
>
> > Limited exploration of how results generalise beyond the specific test cases
>
> > Could you elaborate on how insights from polysemous word evaluation might generalise to other aspects of SAE quality?
>
> The primary message of our work is that existing metrics, such as reconstruction loss or L0-norm, while useful for SAE evaluation, may fail to effectively assess the semantic quality of the extracted features.
>
> For example, in Figure 4 methods like TopK and jumpReLU, developed with a focus on achieving MSE-L0 Pareto optimality, do not necessarily perform well under PS-Eval. Furthermore, as shown in Figure 5, improvements in the PS-Eval metric, such as Specificity, are observed even when MSE and L0-norm metrics become worse.
>
> This demonstrates the importance of complementing existing metrics with semantically focused evaluations like PS-Eval to provide a more comprehensive assessment of SAE quality. Rather than claiming that PS-Eval generalizes to other aspects of SAE quality, we propose it as an additional evaluation axis, enabling the measurement of previously overlooked dimensions and complementing existing metrics for a holistic evaluation framework.
>
>
> > Need for clearer discussion of how PS-Eval relates to existing SAE objectives
>
> As we mentioned so far, several existing objectives for SAEs have been proposed. During training, the primary objective functions typically focus on reconstruction accuracy and sparsity, serving as proxies for the broader expectations of SAE features. However, the evaluation of SAE features remains elusive due to the absence of ground-truth for these features.
>
> To overcome this, an evaluation metric has been proposed using the IOI task in GPT-2, where the ground-truth circuit is known. While this provides insights, it is limited to models with known internal circuits and focuses on proper nouns (e.g., John, Mary) specific to the IOI task. As such, it does not address the evaluation of semantic representations, which are key to what is expected from SAE features.
>
> PS-Eval, in relation to existing SAE objectives, introduces a new perspective by leveraging polysemous words to generate ground-truth representations. This enables the evaluation of semantic representations, filling a critical gap that previous metrics could not address.

---

> ### Author Response · Authors · 2024-11-21
> **Author Response (3/4)**
>
> **Presentation issues**
>
> > Some key terms need clearer definitions, particularly in the introduction (e.g. "semantics-focused evaluation", "polysemous words")
>
> To address the concern regarding the need for clearer definitions of key terms, particularly in the introduction, we have included definitions for polysemantic/monosemantic and polysemous/monosemous in the footer of the introduction.
> Specifically, we define polysemantic/monosemantic as referring to multiple/single functionalities in the activations of LLMs, following the usage in [1]. Meanwhile, polysemous/monosemous refers to multiple/single meanings in words. For example, the word “space” is a polysemous word because it can mean "universe" or "blank area," among other meanings.
> We believe these clarifications will make the terminology more accessible to readers, and please let us know if the unclear points remain.
> > Several figures would benefit from more detailed captions
>
> Thank you for pointing this out. We revised the manuscript to provide more detailed captions for the figures:
>
> For **Figure 3 (left**), we clarified the caption to specify the direction on the x-axis that indicates improvement, making the metric's interpretation more intuitive.
> In **Figure 4**, we added an explanation of the error bars in the caption to ensure clarity regarding their representation.
> For **Figure 5**, we included arrows on both the x-axis and y-axis to indicate the directions corresponding to better performance, along with updated captions to reflect this enhancement.
>
> If there are any other areas that remain unclear or could be further improved, please let us know.
>
>
> > Some sections (particularly the transformer component analysis) need more thorough explanation
>
> To address the concerns from the lack of explanation, we added a diagram in **Appendix H** to clarify which transformer components (e.g., residual, MLP, attention) are being referred to.
>
> Please let us know if some specific sections or points need additional explanation.
>
> > **Quite a few typos**
>
> Thank you for pointing out these issues. We revised the manuscript to correct all identified typos and incomplete sentences, ensuring clarity and accuracy throughout.

---

> ### Author Response · Authors · 2024-11-21
> **Author Response (4/4)**
>
> **Q3**
>
> > The logit lens analysis shows promising results - could you expand on how this validates your evaluation method?
>
>
> Based on the feedback, we positioned the logit lens results to validate our evaluation method, PS-Eval, and moved this discussion to **Section 5** for greater clarity. The logit lens analysis confirms that SAE features correspond to word meanings, providing strong evidence for the validity of our approach.
>
> Additionally, we are exploring ways to extend this qualitative analysis into a more quantitative evaluation. One potential approach is leveraging automated evaluation methods using LLMs, as demonstrated in works like [1].
> To illustrate this, we conducted a preliminary experiment using the results from Table 3 of our paper. Specifically, we focused on the example involving the word "space". In this case, context 1 refers to "space" in the sense of "universe" or "outer space," while context 2 refers to "space" as in "blank space" or "gap."
> We provided the following input to GPT-4:
>
> Input example:
> ```
> ## Instruction:
> You will be provided with two contexts containing the same target word. Additionally, related words for the target word will also be given.
>
> ## Task:
> 1. Determine whether the target word has the same meaning or different meanings in these two contexts.
> 2. Explain the meaning of the target word in each context.
>
> ## Output Format:
> - Same or Different: [Your answer]
> - Meaning in Context 1: [Your explanation]
> - Meaning in Context 2: [Your explanation]
>
> ## Context1 related words
> flight, plane, shuttle, gravity, craft, Engineers, planes
>
> ## Context2 related words
> Layout, occupied, spaces, vacated, space, shuttle, occupancy
> ```
>
> Output example:
> ```
> Output:
> Same or Different: Different
> Meaning in Context 1: The target word refers to a vehicle or object designed for travel or transportation in outer space, such as a spacecraft or shuttle.
> Meaning in Context 2: The target word refers to a physical area or expanse that can be occupied or left empty, often used in the context of spatial arrangements or occupancy.
> ```
>
> The results suggest that GPT-4 can interpret the logit lens outputs from the SAE features in a manner consistent with human intuition. Specifically, it can distinguish whether a feature corresponds to the same or different meanings. Additionally, its reasoning aligns with plausible interpretations, such as identifying that context 1 refers to "universe" or "outer space" while context 2 refers to "blank space".
>
> Using an LLM(GPT-4o)  as an evaluator opens new possibilities for extending the implications of the logit lens results. By systematically applying such methods, we can quantitatively validate the alignment of SAE features with human interpretations, addressing your question in a rigorous and scalable manner.
>
> We have included these results in **Appendix N**.
>
> - [1] Cunningham et al., Sparse Autoencoders Find Highly Interpretable Features in Language Models. https://arxiv.org/abs/2309.08600
>
> Thank you for the detailed feedback. We trust our responses address the points raised, and we would be happy to provide additional clarification if required.

---

> ### Author Response · Authors · 2024-11-26
> **A Reminder to Reviewer fi7z**
>
> Again, Thank you for your valuable review and constructive feedback on our paper. As we have not yet received a response, we would like to kindly remind Reviewer fi7z to review these revisions. In the response thread, we have thoroughly responded to your core conceptual concerns. Additionally, we have revised the structure of the paper to improve clarity and have expanded discussions on the scope of our research, incorporating Related works and detailed explanations in the appendix.
>
> We would sincerely appreciate it if you could kindly assess whether our revisions have effectively addressed your concerns.

---

> > ### Comment · Reviewer_fi7z · 2024-11-26
> >
> > I have reviewed your responses carefully and find that they substantially strengthen the paper's contribution. Your treatment of the core theoretical concern - that SAE features should map directly to word meanings - is now well-supported through multiple lines of evidence. The combination of prior literature showing specific correlations, targeted experimental design, and validation through logit lens results builds a convincing case for this fundamental assumption. This addresses what was previously the paper's most significant weakness.
> >
> > The repositioning of PS-Eval as a complementary metric rather than a wholesale replacement for existing evaluation methods is more measured and realistic. This framing better serves the paper's contribution while avoiding overreach. The explicit connections drawn between PS-Eval and established metrics like MSE and L0-norm help readers understand where this work fits in the broader landscape of SAE evaluation. However, I would still like to see more exploration of cases where PS-Eval might give misleading results or break down.
> >
> > The methodological presentation has improved markedly. The addition of edge cases and failure modes documentation provides necessary context for practitioners, though I believe this material deserves promotion from the appendix to the main text. The clarified terminology and improved figure captions make the technical content more accessible without sacrificing rigor. The new diagram explaining transformer component analysis fills what was a significant gap in the methodology section.
> >
> > Your introduction of GPT-4 evaluation for logit lens outputs is an interesting development that deserves more prominence. This approach offers a potential path toward more systematic validation of SAE feature interpretation. While preliminary, these results suggest promising directions for future work. I would encourage expanding this analysis in the main text rather than relegating it to an appendix.
> >
> > Based on these improvements, I am raising my score from 5 to 6. The paper now makes a meaningful contribution to SAE evaluation methodology, with clear theoretical foundations and practical utility. While some limitations remain - particularly in the breadth of validation and the full exploration of edge cases - the core contribution is sound and valuable. For the camera-ready version, I recommend bringing more of the supporting material from the appendices into the main text, particularly the edge cases discussion and GPT-4 evaluation results. This would strengthen the paper's immediate impact on practitioners while maintaining its theoretical rigor.

---

> > > ### Author Response · Authors · 2024-11-26
> > >
> > > Thank you for your thoughtful and constructive feedback.
> > >
> > > - As suggested by the reviewer, we are working on extending the quantitative evaluation of the Logit Lens using GPT-4. While it is uncertain whether we can include these updates given the approaching PDF modification deadline, we will make every effort to incorporate the results into the final version.
> > > - Regarding the discussion on failure modes, we anticipate deriving numerous concrete examples from the extended quantitative evaluation of the Logit Lens. Despite space constraints, we aim to include as many insights as possible in the camera-ready version to highlight important considerations for practitioners.
> > >
> > > We sincerely appreciate the reviewers' feedback, which has greatly helped refine the contributions, positioning, and limitations of our paper.

---

### Official Review · Reviewer_Nb2h · 2024-11-04

**Soundness:** 3
**Presentation:** 2
**Contribution:** 3
**Rating:** 5
**Confidence:** 4

**Summary:**

This paper proposes a new interpretability metric to substitute the origin reconstruction and sparsity loss in sparse autoencoders. This metric evaluates whether the words of two meanings are the same in features and vice versa. This paper has two findings with this metric: 1. the objective of maximizing reconstruction and sparsity loss does not increase this metric. 2. high layers and attention modules play an essential role in modeling specific semantics of words.

**Strengths:**

1. This paper first examines the ability to identify polysemantics in the interpreter tool of sparse autoencoders.

2. This paper provides a tool to inspect semantic specificity in the transformers.

**Weaknesses:**

This paper lacks several justifications. Refer to the Questions.

**Questions:**

1. What is the bar for the features to be the same and different? How do you tune this bar?

2. Why do late or large language models perform worse in identifying semantic similarity, although they got significantly higher scores on challenging benchmarks?  The performance of this metric does not align with the performance on benchmarks. Could you please discuss potential reasons for this misalignment between their metric and benchmark performance or provide additional analysis exploring this phenomenon?

3. In your setting, recall and specificity are more valuable than other metrics.

4. Why do we need to calculate this metric on the representations of the intermediate layer of SAE  but not AE or the origin representations? Can you provide a comparison?

5. Which layer of the transformer do you test as default?

6. Does this metric generalize better to other interpretability than the origin reconstruction and sparsity loss?

---Post Rebuttal Review---
I have had a discussion with the authors; I think the authors have clarified several issues and promise to add more details and comparisons in the revised version. Thus, I decided to raise the score from 3 to 5.

---

> ### Author Response · Authors · 2024-11-21
> **Author Response (1/2)**
>
> We thank the reviewer for the careful reading and feedback. Please let us know if our responses in the following address your concerns.
>
> We revised the paper based on the reviewers’ comments, and the major edit was highlighted with coloring (purple). Please also check the updated manuscript.
>
> **Q2**
>
> > Why do late or large language models perform worse in identifying semantic similarity, although they got significantly higher scores on challenging benchmarks?
>
>
> First of all, we would like to emphasize that the evaluation target of the proposed metric in this paper is the **Sparse Autoencoder (SAE)**, which maps the internal activations of LLMs into interpretable spaces Cunningham et al., [1], **not LLMs themselves**. While previous research (Gao et al., Templeton et al., )[2,3] has used proxy metrics such as reconstruction error and L0 norm to evaluate SAEs, this paper introduces a novel metric, PS-eval, to assess whether the features learned by SAEs capture monosemantic features.
>
> Thus, **a low PS-Eval score in this paper does not imply that LLMs perform worse in identifying semantic similarity**. Rather, this work focused on proposing **a new evaluation protocol for SAEs as interpretability tools for LLMs**, which has been missed in the previous literature. We did not focus on evaluating the semantic capabilities of the LLMs themselves.
>
> - [1] Cunningham et al., Sparse Autoencoders Find Highly Interpretable Features in Language Models. https://arxiv.org/abs/2309.08600
> - [2] Gao et al.,Scaling and evaluating sparse autoencoders.
> https://arxiv.org/abs/2406.04093
> - [3] Templeton et al., Scaling Monosemanticity: Extracting Interpretable Features from Claude 3 Sonnet. https://transformer-circuits.pub/2024/scaling-monosemanticity/
>
> **Q6**
>
> > Does this metric generalize better to other interpretability than the origin reconstruction and sparsity loss?
>
> The original reconstruction and sparsity loss functions are objectives used for training SAEs, but **they do not directly measure interpretability in a word-semantic space**, rather evaluating the compressibility. What we aim to achieve with SAEs is to map the polysemantic and difficult-to-interpret activations of LLMs into monosemantic features. To evaluate this specific capability, we propose our new evaluation pipeline. Therefore, we believe that this metric generalizes better to other interpretability than the origin reconstruction and sparsity loss.
>
> **Q4**
>
> > Why do we need to calculate this metric on the representations of the intermediate layer of SAE but not AE or the origin representations? Can you provide a comparison?
>
> **Figure 3** in the original manuscript compares the similarity of original activations and SAE features when words with different meanings are presented. Since the context introduces different meanings, the similarity between these activations and SAE features should ideally decrease (shifting the distribution to the right).
>
> The results show that the distribution of SAE features is shifted further to the right compared to LLM activations, indicating that **SAE features better capture the distinction between different meanings than original polysemantic activations in LLMs**.
>
> Additionally, we conducted a comparison between SAEs and dense autoencoders (trained without sparsity $\mathrm{L}_{0}$ loss) in **Appendix I**, which demonstrates that SAEs (with sparsity constraints) outperform fully dense autoencoders in terms of interpretability of polysemous words. We believe these support the necessity of SAE for the fine-grained analysis, and emphasize the importance of our proposed evaluations.
>
> **Q3**
>
> > In your setting, recall and specificity are more valuable than other metrics.
>
> We believe that all of those (accuracy, precision, recall, f1, specificity) can contribute to characterizing and analyzing the behavior of learned SAE, and must be high if SAE has optimal features. While our PS-Eval suggests the holistic evaluation with all the metrics, we also note the careful usage of some metrics, such as specificity, when singly optimizing them.
>
> For example, if the learned feature is random, the similarity of the SAE feature is always low, and then the specificity results in a meaninglessly high value, which may not reflect the quality of SAE features.
>
> In the revised paper, we clarify these and added the recommendation of the metrics in **Section 4.2**.
>
> Moreover, to ensure a comprehensive evaluation, we systematically assessed all relevant metrics, including recall and specificity. The detailed results and analysis can be found in **Appendix G**.
>
>
>
>
> **Q1**
>
> > What is the bar for the features to be the same and different? How do you tune this bar?
>
> As described in Section 4.2, our paper defines that, (1) the SAE features are the same if the index of the highest activation exactly matches, and (2) the SAE features are different if the index of the highest activation does not match.

---

> ### Author Response · Authors · 2024-11-21
> **Author Response (2/2)**
>
> **Q5**
>
> > Which layer of the transformer do you test as default?
>
> We use **layer 4** as the default layer. As shown in Figure 5, the metric begins to improve around layer 4. That is why we have selected it as the default layer for our experiments. To make this clearer, we added a detailed explanation in **Section 3.2**.
>
> We sincerely appreciate the reviewer’s insightful feedback. Please let us know if our responses resolve your concerns or if additional clarification is required.

---

> > ### Author Response · Authors · 2024-11-26
> > **A Reminder to Reviewer Nb2h**
> >
> > Thank you again for your valuable review and constructive feedback on our paper. As we have not yet received a response, we would like to kindly remind Reviewer Nb2h that we have thoroughly addressed your concerns. Specifically, we have clarified our motivation and conducted additional experiments involving dense SAE. These updates have been incorporated into the revised paper, which we believe has been significantly strengthened as a result.
> >
> > We would greatly appreciate it if you could review these updates and kindly reconsider your score in light of our revisions to address your comments.

---

> ### Comment · Reviewer_Nb2h · 2024-11-26
> **Reply to rebuttal**
>
> After reading your rebuttal and discussion with other reviewers, I think you have answered Q1, Q3, and Q4.
>
> For Q5, in section 3.2, you write that layers after 6 show an increasing trend in specificity, but not layer 4. Moreover, why do we choose the layer that starts increasing the metric but does not have the highest metric?
>
> For Q2 and Q6, the authors emphasized the scope of the paper but did not directly answer my questions. For Q2, please explain why late or large LLMs have lower PS-eval scores.  For Q6, this paper does not test whether optimizing this metric determines another interparty metric.

---

> ### Author Response · Authors · 2024-11-27
> **Author Reply (1/3)**
>
> We appreciate the response from the reviewers. We will answer your remaining questions as follows.
>
>
> **> Re: Q5. The choice of the layer**
>
> In our cases, we found that, from the 4th layer, the preliminary experiments of the logit lens (similar to Section 5) have started to show distinguishable results among the polysemous words. This is why we selected the 4th layer as default.
>
> We believe that any choice of the LLM layer for SAE training could be accepted in the community. In the previous literature, there was no agreeable criterion of how we could choose the index of layers based on some metrics. For instance, Cunningham et al., [1] use the 11th layer of Pythia-410M in Figure 3 of their paper, and Gao et al., [2] use a layer 5/6 of the way for GPT-4 and a layer 3/4 of the way for GPT-2-small, but those were arbitrary and those papers did not have rigorous justification for the choice based on the quantitative criterion.
>
> Following Cunningham et al., [1], in addition to the main results from SAEs trained with the 4th layer,  we also included results from SAEs trained with all other layers in **Figure 12**. Moreover, we think that the analysis/consequence of our paper does not significantly differ in case we use another layer. To verify this, we are planning to add results from SAEs with the 6th layer. Due to the time constraint, we cannot include the results in the main PDF at this time, but we are able to follow up on the results here in a markdown format in the next few days, and will surely include them in a final version.
>
> - [1] Cunningham et al., Sparse Autoencoders Find Highly Interpretable Features in Language Models. 2023. https://arxiv.org/abs/2309.08600
>
> - [2] Gao et al., Scaling and evaluating sparse autoencoders. 2024. https://arxiv.org/abs/2406.04093

---

> ### Author Response · Authors · 2024-11-27
> **Author Reply (2/3)**
>
> **> Re: Q2. Performance of open SAEs from the latest LLMs in Section 8 (Table 4)**
>
> We apologize for our misunderstanding about your question; we thought that your question was about LLMs in general, rather than about the open SAE results we presented in Section 8 (Table 4).
>
> In our paper, we extensively evaluate SAEs from GPT-2-small and also assess the quality of open SAEs from GPT-2-small, Pythia 70M, and Gemma2-2B (Section 8, Table 4). The results imply that open SAE trained from larger/more recent LLMs does not always achieve better performance on PS-Eval (such as GPT-2-small > Gemma2-2B). Here, we would like to clarify the reasons for the performance gap.
>
> **1. It is non-trivial whether more capable LLMs have more interpretable intermediate activations or not.**
>
> Bereska et al., [1] mentioned that whether larger and more recent LLMs produce activation patterns that are more interpretable from the perspective of SAE remains unclear despite their better capability on natural language tasks. Capable/large-scale LLMs may have more complex activation internally. Our results in Section 8 are aligned with their statement; open SAEs from Pythia 70M or Gemma2-2B are not always better than that of GPT-2-small in PS-Eval. In principle, we should assume that the performances of LLMs and SAEs are independent. It is not obvious (and examined so far) whether the interpretability of SAE correlates with the capability of base LLMs, and we think it is worth investigating in future works. We added this point to **Appendix M** (Discussion and Limitation) as a part of the future direction.
>
> **2. Optimizing $\mathrm{L}_{0}$-MSE Pareto frontier does not always lead to better interpretability.**
>
> Open SAE from more recent LLMs has incorporated several techniques to improve the L0-MSE Pareto frontier.
> For instance, open SAE from Gemma2-2B leverages JumpReLU as an activation function. However, as we showed in Section 6.3, such a method to improve the L0-MSE Pareto frontier does not always improve the performance of interpretability; in Figure 4, the standard ReLU function achieves better F1 scores than JumpReLU, that can realize better L0-MSE Pareto frontier. Since open SAEs based on later LLMs are designed to achieve a better Pareto frontier, this affects the performance of PS-Eval.
>
>
> **3. The performance of SAE may depend on architectures and hyperparameters specific to base LLMs.**
>
> As we showed in the experiment sections (Figures 3 and 5), the tradeoff of the performances among L0, MSE, and PS-Eval might rely on the architectural and hyperparameter choices, such as expand ratio, activation, and the index of layers. In the experiments, we evaluated the off-the-shelf open SAEs that were tailored for L0-MSE, but the optimal architectures and hyperparameters in terms of PS-Eval can differ from those (due to the severe computation and cost constraints, it was not feasible for us to train all of those SAEs from scratch). We have discussed this point in Section 8 since before.
>
> We believe our empirical results in Section 8 are appropriately supported by the previous literature and observations.
>
> - [1] Bereska et al., Mechanistic Interpretability for AI Safety - A Review, TMLR. 2024. https://openreview.net/forum?id=ePUVetPKu6

---

> > ### Author Response · Authors · 2024-11-27
> > **Author Reply (3/3)**
> >
> > **> Re: Q6. PS-Eval and other metrics**
> >
> > For this point, we would like to clarify that, to the best of our knowledge, previous works have only employed L0
> > (L1) and MSE as proxies of the interpretability metric to evaluate the quality of SAEs [1,2,3,4]. Such a lack of qualitative interpretability metrics (especially focusing on the semantics) for the evaluation in the community was our primal motivation to develop PS-Eval. We have described this in Abstract & Section 1.
> >
> > From this perspective, we have extensively studied the tradeoff between L0/MSE and metrics in PS-Eval (accuracy, F1 score, recall, precision, specificity, etc.) in Figure 3, Figure 5, Appendix F, and Appendix G. Those 2D plots demonstrated that some metrics (e.g., Specificity in Appendix G) have increased as L0/MSE increased, and other metrics (e.g., Accuracy in Figure 3) have decreased as L0/MSE decreased. Based on these observations, we position our PS-Eval as metrics complemented to the existing L0/MSE (rather than the replacement), while expanding the axis for multi-objective assessment. Through the discussion period, we tuned the tone of the revised paper appropriately (e.g., Section 4.2).
> >
> > Moreover, we would like to emphasize that the metrics in PS-Eval are not designed to optimize them individually, rather we recommend improving all of these metrics simultaneously while balancing the metrics tradeoff. PS-Eval plays a critical role when we improve the quality of SAEs through holistic evaluation.
> >
> > By following previous literature on the research of SAE, we believe we do not have mandatory "baselines" to compare except for L0 and MSE. If the reviewer has other interpretability metrics for the evaluation of SAEs in your mind, please feel free to point them out specifically. We will strive to deal with them as far as the window of author response allows, and at least we are happy to include the relevant discussion for clarification.
> >
> > - [1] Templeton et al., Scaling Monosemanticity: Extracting Interpretable Features from Claude 3 Sonnet. 2024. https://transformer-circuits.pub/2024/scaling-monosemanticity/index.html
> >
> > - [2] Gao et al., Scaling and evaluating sparse autoencoders. 2024. https://arxiv.org/abs/2406.04093
> >
> > - [3] Rajamanoharan et al., Jumping Ahead: Improving Reconstruction Fidelity with JumpReLU Sparse Autoencoders. 2024. https://arxiv.org/abs/2407.14435
> >
> > - [4] Taggart. ProLU: A Nonlinearity for Sparse Autoencoders. 2024. https://www.alignmentforum.org/posts/HEpufTdakGTTKgoYF/prolu-a-nonlinearity-for-sparse-autoencoders
> >
> >
> >
> > We thank you again for your active engagement in the discussion process. We hope our additional responses can address your remaining concerns. Please feel free to let us know if you have any extra questions.

---

> > > ### Author Response · Authors · 2024-11-30
> > >
> > > **> Re: Q5. The choice of the layer**
> > >
> > > To supplement the discussion about layer selection, we conducted additional experiments as described in **Section 6**, using the 6th layer as an alternative input activations. The results of these experiments are summarized below.
> > >
> > >
> > > **Figure 3: Accuracy vs. Expand Ratio**
> > >
> > > | Expand Ratio | 8      | 16     | 32     | 64     | 128    |
> > > |--------------|--------|--------|--------|--------|--------|
> > > | Layer 4      | 0.6005 | 0.6328 | 0.6585 | 0.6669 | 0.6576 |
> > > | Layer 6      | 0.4982 | 0.6203 | 0.6370 | 0.6550 | 0.6686 |
> > >
> > > As shown in Figure 3, while the accuracy improves consistently with larger expand ratios, the trends across layers remain broadly similar. This indicates that the choice of the layer does not significantly impact the observed patterns, supporting the robustness of our findings.
> > >
> > >
> > > **Figure 4: Performance Across Activation Functions**
> > >
> > > **F1 Score**
> > >
> > > | Method      | ReLU   | STE(jump=0.001) | STE(jump=0.0001) | topk=384 | topk=96 | topk=192 |
> > > |-------------|--------|-----------------|------------------|----------|---------|----------|
> > > | Layer 4     | 0.7623 | 0.6667          | 0.6590           | 0.6550   | 0.6490  | 0.6166   |
> > > | Layer 6     | 0.7509 | 0.6203          | 0.6136           | 0.5985   | 0.6203  | 0.6316   |
> > >
> > > **Precision**
> > >
> > > | Method      | topk=384 | ReLU   | topk=192 | topk=96 | STE(jump=0.001) | STE(jump=0.0001) |
> > > |-------------|----------|--------|----------|---------|-----------------|------------------|
> > > | Layer 4     | 0.6939   | 0.6593 | 0.5218   | 0.5251  | 0.5000          | 0.4964           |
> > > | Layer 6     | 0.5170   | 0.6729 | 0.5178   | 0.5136  | 0.4901          | 0.4599           |
> > >
> > > **Recall**
> > >
> > > | Method      | STE(jump=0.001) | STE(jump=0.0001) | ReLU   | topk=96 | topk=192 | topk=384 |
> > > |-------------|-----------------|------------------|--------|---------|----------|----------|
> > > | Layer 4     | 0.9951          | 0.9802           | 0.9034 | 0.8813  | 0.7536   | 0.6203   |
> > > | Layer 6     | 0.9501          | 0.8615           | 0.8094 | 0.8523  | 0.8094   | 0.7104   |
> > >
> > > The results in Figure 4 show that, regardless of the layer used, ReLU consistently achieves the highest F1 score, aligning with findings from Layer 4. Additionally, the general trends across activation functions are not significantly affected by the choice of the layer, confirming that our conclusions hold broadly.
> > >
> > > These findings align with prior work, where arbitrary layer choices have been commonly accepted.
> > > We will include the results from this comparison in the **Appendix** of the final version for transparency and completeness.

---

> > > > ### Comment · Reviewer_Nb2h · 2024-12-03
> > > > **Response to rebuttal2**
> > > >
> > > > I have increased the score from 3 to 5; I recommend you add more details and comparisons in the revised version.

---

> > > > > ### Author Response · Authors · 2024-12-03
> > > > >
> > > > > Thank you for your valuable feedback. We appreciate your recommendation and will include all the experiments and comparisons discussed during the Discussion Period in the revised version of the paper.

---

### Official Review · Reviewer_s56H · 2024-11-05

**Soundness:** 3
**Presentation:** 3
**Contribution:** 3
**Rating:** 6
**Confidence:** 4

**Summary:**

This paper introduces a new framework, called PS-Eval, for evaluating Sparse Autoencoders (SAEs), which are widely used to interpret LLMs. The framework repurposes the WiC dataset (Pilehvar & Camacho-Collados, 2019), originally designed as a binary classification task. Specifically, the paper proposes using a subset of WiC to evaluate the extent to which the features learned by SAEs are monosemantic (the ideal case for SAEs). With this dataset, the authors suggest evaluating SAEs using metrics like precision, recall, and specificity. The paper provides a detailed analysis of SAE performance across different layers and transformer components (e.g., residual, MLP, and attention) and compares activation functions such as ReLU, TopK, and JumpReLU.

**Strengths:**

I really enjoyed reading this paper, as it addresses a well-known issue with SAEs that has been overlooked for a long time. It does so by introducing a context-sensitive metric for monosemantic representation, making the paper timely and addressing a current gap in interpretability research. I also appreciated the depth of the experiments, which cover different transformer modules and layers. The analysis is insightful, clearly demonstrating that standard metrics (MSE and sparsity ratio) can be insufficient for capturing monosemantic features. I believe such analyses are very valuable for future SAE design and application. Overall, the paper is well-organized, well-motivated, easy to read, and offers insightful conclusions.

**Weaknesses:**

Even though I enjoyed reading this paper, my main issue is that there is no baseline for comparison. For example, in Table 4, it is hard to conclude which values of accuracy/precision/recall/specificity signify a good SAE. In other words, even with this new framework, it is unclear whether these SAEs can be trusted. Having a good baseline (e.g., a random SAE or fully dense SAE) could help clarify this issue.

**Questions:**

Can the PS-Eval framework be extended to other models or applications outside of NLP, or is it fundamentally tied to the context-sensitive nature of language?

Figure 3: Could you explain why LLM activations are considered less interpretable than SAE features in Figure 3? For me, the fact that LLM activations are squished into the [0, 0.2] interval doesn't make them less interpretable. Instead, it seems like a matter of finding a different cutoff value for LLM activations. Also, how many bins were used for the plot?

---

Minor comments:

In line 316, you say, "Specificity measures how often the different meaning words have the same SAE features." The correct wording should be "different SAE features," as per Table 2.

In line 96: "In our evaluation across different Transformer components, specifically the residual layers, MLPs, and self-attention." I believe a part of the sentence is missing.

---

> ### Author Response · Authors · 2024-11-21
> **Author Response (1/2)**
>
> We thank the reviewer for the constructive feedback. Please let us know if our responses in the following address your concerns.
>
> We revised the paper based on the reviewers’ comments, and the major edit was highlighted with coloring (purple). Please also check the updated manuscript.
>
> **W1**
>
> > Which metrics (accuracy/precision/recall/specificity) signify a good SAE? Having a good baseline (e.g., a random SAE or fully dense SAE) could help clarify this.
>
> We believe that all of those (accuracy, precision, recall, f1, specificity) should be high if SAE has optimal features. However, we also agree that, while some metrics still can characterize and help analyze the behavior of learned SAE, even some trivial baselines may achieve high values.
>
> To clarify this, we prepared the following variants you suggested:
>
> 1. **Random SAE Baseline:**
>
> In **Figure 16, 17** in the revised paper (Appendix J), we included results from random SAE where its features are randomly activated (a black dashed line). Standard SAE significantly outperforms random SAE, demonstrating that SAE achieves higher accuracy in classifying the polysemous words and learns interpretable features.
>
> In **Figure 18, 19** in the revised paper, we included results from random SAE. The high Specificity observed in the random SAE.  This occurs because, in the case of a random SAE with feature dimensions as large as 24,576, most activations are inherently distinct, resulting in a high Specificity baseline for random SAE. This highlights that Specificity is more suitable for evaluating the characteristics of SAE rather than serving as the sole optimization target.
>
> 2. **Fully Dense SAE Baseline:**
>
> Additionally, in **Appendix I**, we added results from a fully dense SAE (by removing sparsity constraints from the objective). The comparison shows that fully dense SAEs perform worse than standard SAE (with sparsity constraints) in terms of accuracy in classifying the polysemous words．
>
> Therefore, we basically recommend a holistic evaluation with all metrics, and, in case you would like to optimize one of them independently, we recommend the usage of class-balanced metrics such as accuracy or F1 scores as safer options.
> In the revised paper, we clarified these and added the recommendation of the metrics in **Section 4.2**.
>
> **Q1**
>
> > Can the PS-Eval framework be extended to other models or applications outside of NLP, or is it fundamentally tied to the context-sensitive nature of language?
>
>
> While the SAE was originally proposed as a tool to analyze polysemantic activations in language models, it has recently been applied to other domains, such as diffusion models and vision-language models like CLIP [1, 2].
>
> For example, Surkov et al.,[1] discusses the application of SAE to the Stable Diffusion (text-to-image) model. Since this model relies on text inputs, it is plausible to evaluate its behavior using PS-eval-like methods to analyze responses to polysemous text inputs.
> In the case of vision-based models like CLIP Sangwu Lee.,[2], we think it is possible to extend the PS-Eval framework by utilizing polysemous image inputs — images that can be interpreted in multiple ways. For example, Shahgir et al. [3] provides an optical illusion dataset specifically designed for vision-language models, which could serve as a useful reference for developing a PS-Eval framework tailored to image inputs.
>
> These examples suggest that the PS-Eval framework has the potential to be extended beyond NLP to other domains, provided that the context or input ambiguity relevant to the application is appropriately addressed.
>
> In the revision, we expanded on this point in the Discussion and Limitation section of **Appendix L**, where we discussed the potential for extending the PS-Eval framework beyond NLP to other domains.
>
>
> - [1] Surkov et al., Unpacking SDXL Turbo: Interpreting Text-to-Image Models with Sparse Autoencoder. https://arxiv.org/abs/2410.22366
> - [2] Sangwu Lee., Gazing in the Latent Space with Sparse Autoencoders, https://re-n-y.github.io/devlog/rambling/sae/
> - [3] Shahgir et al., IllusionVQA: A Challenging Optical Illusion Dataset for Vision Language Models. https://arxiv.org/abs/2403.15952

---

> ### Author Response · Authors · 2024-11-21
> **Author Response (2/2)**
>
> **Q2**
>
> > Figure 3: Could you explain why LLM activations are considered less interpretable than SAE features in Figure 3?
>
> Figure 3 compares the cosine distances of LLM activations and the cosine distances of  SAE features when different polysemous contexts (poly-contexts) are input. Since these contexts have different meanings, it is preferable for their corresponding representations to be farther apart, meaning a higher value on the x-axis (toward the right) indicates better performance in distinguishing meanings.
>
> To make this metric clearer, we introduced the term *Polysemous Distinction* (please refer to Section 6.1) and added arrows and captions to Figure 3 to guide interpretation. The results show that SAE features (**green**) exhibit greater cosine distances compared to LLM activations (**blue**), suggesting that SAE features are better at distinguishing polysemantic contexts, which supports our claim of improved interpretability.
>
> If there are other aspects of Figure 3 or our explanation that remain unclear, please let us know, and we would be happy to clarify further.
>
> **Minor1**
>
> > In line 316, you say, "Specificity measures how often the different meaning words have the same SAE features." The correct wording should be "different SAE features," as per Table 2.
>
> Thank you for pointing this out. It was a typo. We have corrected it in the revised version.
>
>
> **Minor2**
>
> > In our evaluation across different Transformer components, specifically the residual layers, MLPs, and self-attention.
>
> In the updated manuscript, the sentence was revised for clarity:
> "We evaluated different Transformer components: residual layers, MLPs, and self-attention."
>
>
> Again, we appreciate the detailed comments from the reviewer. Please let us know if our responses address your concerns. If needed, we will be happy to clarify them further.

---

> > ### Comment · Reviewer_s56H · 2024-11-22
> > **Comment**
> >
> > Thank you for the extra experiments and clarifications. I'll increase my soundness score accordingly.

---

> > > ### Author Response · Authors · 2024-11-26
> > >
> > > Thank you for reviewing our response.
> > > If you have any additional questions, we would be happy to address them within the remaining timeline.
> > >
> > > We sincerely appreciate your engagement and effort given the tight schedule.

---

### Author Response · Authors · 2024-11-21
**Summary of Revision in Author Response (1/2)**

We appricate detailed reading and suggestive feedback from all the reviewers. We revised the paper based on the reviewers’ comments, and **the major edit was highlighted with coloring (purple).**

The key changes is summarized below:

- Added a **random baseline** in Figures 17, 18, 19, and 20 in Appendix J (Reviewer s56H).
- Included a **fully dense baseline** in Appendix I (Reviewer s56H, Nb2h).
- Updated **Figure 3 captions** (Reviewers s56H, Nb2h, and  fi7z).
- Moved the **logit lens results** to Section 5 (Reviewer fi7z).
- Expanded **Section 2 (Related Work)** to clarify the scope of our metrics (Reviewer fi7z).
- Added a **Discussion and Limitation** section in Appendix M (Reviewers s56H, fi7z).
- Explicitly defined the **default layer** in Section 3.2 (Reviewer Nb2h).
- Added experiments for smaller **Top-k values** and **STE jump ReLU** to Appendix K (Reviewer k8GD).
- Updated **Figure 3** by changing "normalized $\mathrm{L}_0$" to simply $\mathrm{L}_0$ (Reviewer k8GD).
- Included experiments on the **number of dead latents** and the effect of ghost gradients in Appendix A (Reviewer k8GD).
- Added **$\mathrm{L}_0$ results for each activation function** in Appendix L (Reviewer k8GD).
- Added a figure illustrating the **conponents of the Transformer architecture** (Residual, MLP, Attention) in Appendix H (Reviewer fi7z).
- Revised terminology: Defined **semantic** and **polysemous** in the introduction's footer to clarify their use in the abstract and introduction (Reviewer F17z).
- Corrected typos and minor grammatical issues (Reviewer s56H, fi7z).


## **Clarification on the Validity of Our Assumptions**

Reviewer Fi7z raised the concern that this paper assumes SAE features should directly map to word meanings, suggesting this is a strong assumption that requires further justification.

We justify our assumption on the SAE features from three perspectives:  (1) findings from previous studies, (2) our experimental design, and (3) the results of our logit lens analysis.

**Findings in Prior Works**

It is well-established in prior research that SAE features map to word meanings.
For instance, in Templeton et al., (2024) [1], features corresponding to specific words such as "Golden Gate Bridge" or "Brain Science" were identified.
Similarly, GPT-4 evaluations in Table 1 of Cunningham et al., (2023) [2] demonstrated that SAE features correspond to word meanings such as "legal terms."
More recently, Li et al., (2024) [3] analyzed the geometric structure of concepts in LLMs under the assumption that SAE features encode **semantic concepts**, ranging from specific word meanings such as "king" and "queen" to more abstract concepts like "code" and "math."
These findings align with our underlying assumption that SAE features respond specifically to word meanings.

**Our Experimental Design**

As you noted in your "tweet" example, we acknowledge that, depending on the context, SAE features may fire for elements beyond word meanings. To address this, our experiments are specifically designed to embed word meanings into SAE features.our experimental setup explicitly focused on embedding word meanings into SAE features. As described in Section 4.2, SAE features were trained on activations of the target word within the prompt, such as:

`{context}. The {target_word} means `

This design ensures that the meaning of the target word is directly incorporated into the SAE features. By tailoring our framework in this way, we aimed to align SAE features closely with “word meaning”.

**Results of the Logit Lens Analysis**

Building on the experimental design above, our results further validate that SAE features correspond to word meanings. As shown in Table 3, SAE features fire based on the meanings of words. For example, in the case of the token "space" (Table 3, left column), the feature that activates depends on the context, indicating distinct meanings: "space" as "outer space" (green) and "space" as "blank space" (orange).

While we agree that SAE features may also fire for aspects other than word meanings because we don't know what the actual atomic units of computation in a model are, based on these empirical observations and findings, we also believe that the assumption in question is well-justified.

In response to this concern, we have restructured the manuscript to clarify the validation of our assumption. The logit lens results have been moved to Section 5, where they are now explicitly positioned as evidence supporting the assumption that SAE features map to word meanings. This reorganization highlights the role of these results in demonstrating how SAE features align with word meanings, providing a clearer narrative flow.

---

> ### Author Response · Authors · 2024-11-21
> **Summary of Revision in Author Response (2/2)**
>
> - [1] Scaling Monosemanticity: Extracting Interpretable Features from Claude 3 Sonnet, https://transformer-circuits.pub/2024/scaling-monosemanticity/index.html
> - [2] Cunningham et al., Sparse Autoencoders Find Highly Interpretable Features in Language Models. https://arxiv.org/abs/2309.08600
> - [3] Li et al., The Geometry of Concepts: Sparse Autoencoder Feature Structure. https://arxiv.org/abs/2410.19750

---

> > ### Author Response · Authors · 2024-11-27
> >
> > **General Response (2024/11/27)**
> >
> > From the last update, we further revised the paper based on the discussion with the reviewers.
> >
> > Below, we summarize the key updates:
> >
> > - Add a discussion on the late and larger  LLMs and PS-Eval in **Appendix M.** (Reviewer Nb2h)
> >
> > - Add the description of L0 instead of normalized L0 in **Section 6.2**. (Reviewer k8GD)
> >
> > - Include details on the initialization method, dead latents, and Ghost Grads in **Section 3**. (Reviewer k8GD).
> >
> > - Incorporate the results from Appendix K, including smaller `k` values for TopK and STE JumpReLU, into **Figure 4**. Correspondingly, Update the explanation in **Section 6.3** to reflect these changes (Reviewer k8GD).
> >
> > - Add the L0 results for STE JumpReLU to the **Table 9** in **Appendix L** (Reviewer k8GD).
> >
> > We hope these revisions address the reviewers' concerns and further strengthen the clarity and presentation of our work.

---

### Author Response · Authors · 2024-12-04
**Summary of Discussion (1/3)**

Dear Reviewers, AC, and SAC,

We would like to thank you for your great effort in reviewing our paper.
While this is a last minute before closing discussion period, we believe our author responses have addressed all the raised concerns. Again, we would like to highlight some important discussions and our contributions as follows. It would be great if you take these into consideration for your final decision.


**Discussion and Limitation**

Throughout the discussion period, we have included and updated the Discussion and Limitation section in **Appendix M**, which discusses (1) the possibility of extending PS-Eval into multimodal/text-to-image generation settings (from Reviewer **s56H**), (2) clarification of the context from previous SAE studies (from Reviewer **Nb2h**), and (3) potential failure mode or edge case of our proposed evaluation (from Reviewer **fi7z**).

For the point (2), we discussed that it is unclear that more capable LLMs always lead to more interpretable SAE features, and there is no agreeable criterion of how the index of LLM layers to train SAE. For the point (3), we explained our dependence on maximum activated features rather than the distributional evaluation of SAE features, and the existence of learned SAE features corresponding to the splitted token rather than the meaning in the context (shown in Table 3).

In the final version, to improve the clarity further, **we will move the Discussion and Limitation section into the main text, and move the current Section 8 (analysis on the open SAEs) into the Appendix instead.** We believe this can address the last suggestions from Reviewer **fi7z** and **Nb2h** to add more details and comparisons in the revised version. We also hope this would help the practitioners and readers to understand the scope and limitation of our paper appropriately.

Moreover, considering the limitations and failure cases, we revised each section to make our statements be more measured. Based on the observation that there are tradeoffs between L0/MSE and metrics in PS-Eval, we position our PS-Eval as metrics complemented to the existing L0/MSE (rather than the replacement), while expanding the "semantics" axis among multi-objective assessment for the development of more practical SAEs. In the revised paper, we tuned the tone of discussion appropriately (e.g., **Section 4.2**).

We believe that all the metrics in PS-Eval (accuracy, precision, recall, f1, specificity) can contribute to characterizing and analyzing the behavior of learned SAE, and must be high if SAE has optimal features. While our PS-Eval suggests the holistic evaluation with all the metrics, we also note the careful usage of some metrics, such as specificity, when singly optimizing them. For example, if the learned feature is random, the similarity of the SAE feature is always low, and then the specificity results in a meaninglessly high value, which may not reflect the quality of SAE features. In the revised paper, we clarified these and added the recommendation of the metrics in **Section 4.2**.

---

> ### Author Response · Authors · 2024-12-04
> **Summary of Discussion (2/3)**
>
> **Wide Range of Additional Experiments**
>
> In the discussion period, we have added variety of additional experiments to support our results and claims.
>
> - Updated **Figure 4** has a wide range of comparison among different activation function (different threshold and STE variant for JumpReLU, and different $k$ for TopK; from Reviewer **k8GD**).
> - **Appendix A** has a ablation of Ghost Grads (from Reviewer **k8GD**).
> - **Appendix G** has an extensive comparison among the possible metrics in PS-Eval (Accuracy, Recall, Precision, Specificity, Sensitivity, F1, etc; from Reviewer **Nb2h**).
> - **Appendix I** has a results of dense autoencoder (SAE without L1 regularization) on the PS-Eval as a baseline (from Reviewer **s56H**, **Nb2h**).
> - **Appendix J** has a results of randomly-activated SAE on the PS-Eval as a baseline (from Reviewer **s56H**).
> - **Appendix K**, **Appendix L**, and the follow-up experiments to Reviewer **k8GD** have revealed the relationship among different activation functions in SAEs, PS-Eval, and the L0 sparsity. By incorporating experiments that cover a broader range of $L_0$ sparsity including case where $L_0 < d_\mathrm{model}$, we have made the impact of activation function choices more comprehensive and convincing.  We will appropriately incorporate the follow-up experiments into Appendix K and L.
> - **Appendix N** has a preliminary results towards the automated quantitative evaluation of SAE features, conducted by asking GPT-4 about the similarity of logit lens outcomes (from Reviewer **fi7z**).
>
> Additionally, to address the concern from Reviewer **Nb2h** about arbitrary choices of LLM layer used for SAE training, we compared the performance on PS-Eval between layer 4 (our default) and layer 6 (one of reviewer's suggestion). The results shows that the general performance trends are consistent among different layers, and the choice of the layer does not significantly affect our analysis in the paper.
>
> We will include all the follow-up experiments in Appendix in the revised version. We believe those extensive additional experiments and analysis addressed the last suggestions from the reviewers to add more details and comparisons in the revised version.
>
>
> **Performance of open SAEs from the latest LLMs**
>
> In **Section 8**, we extensively evaluate SAEs from GPT-2-small and also assess the quality of open SAEs from GPT-2-small, Pythia 70M, and Gemma2-2B (Section 8, Table 4). The results imply that open SAE trained from larger/more recent LLMs does not always achieve better performance on PS-Eval (such as GPT-2-small > Gemma2-2B). This is because (1) it is non-trivial whether more capable LLMs have more interpretable intermediate activations or not, (2) optimizing L0-MSE Pareto frontier does not always lead to better interpretability, and (3) the performance of SAE may depend on architectures and hyperparameters specific to base LLMs.
>
> The point (1) is supported by the previous work by Bereska et al., [1]; whether larger and more recent LLMs produce activation patterns that are more interpretable from the perspective of SAE remains unclear despite their better capability on natural language tasks.
>
> For the point (2), **Section 6.3** in our paper revealed that a method to improve the L0-MSE Pareto frontier does not always improve the performance of interpretability on PS-Eval, while open SAEs are developed by optimizing it.
>
> For the point (3), we showed in **Figures 3** and **5** that the tradeoff of the performances among L0, MSE, and PS-Eval might rely on the architectural and hyperparameter choices, such as expand ratio, activation, and the index of layers. The optimal architectures and hyperparameters in terms of PS-Eval can differ from those. We have discussed this point in Section 8 since before.
>
> We believe our empirical results in Section 8 are appropriately supported by the previous literature and observations.
>
> - [1] Bereska et al., Mechanistic Interpretability for AI Safety - A Review, TMLR. 2024. https://openreview.net/forum?id=ePUVetPKu6

---

> > ### Author Response · Authors · 2024-12-04
> > **Summary of Discussion (3/3)**
> >
> > **Choice of the layer when training SAEs**
> >
> > As we mentioned above, we believe that any choice of the LLM layer for SAE training could be accepted in the community. In the previous literature, there was no agreeable criterion of how we could choose the index of layers based on some metrics. For instance, Cunningham et al., [1] use the 11th layer of Pythia-410M in Figure 3 of their paper, and Gao et al., [2] use a layer 5/6 of the way for GPT-4 and a layer 3/4 of the way for GPT-2-small, but those were arbitrary and those papers did not have rigorous justification for the choice based on the quantitative criterion.
> >
> > In our cases, we found that, from the 4th layer, the preliminary experiments of the logit lens have started to show distinguishable results among the polysemous words. This is why we selected the 4th layer as default.
> >
> > Following Cunningham et al., [1], in addition to the main results from SAEs trained with the 4th layer, we also included results from SAEs trained with all other layers in Figure 12. Moreover, through the follow-up experiments with 6th layer, we demonstrate that the analysis/consequence of our paper does not significantly differ in case we use another layer.
> >
> > [1] Cunningham et al., Sparse Autoencoders Find Highly Interpretable Features in Language Models. 2023. https://arxiv.org/abs/2309.08600
> >
> > [2] Gao et al., Scaling and evaluating sparse autoencoders. 2024. https://arxiv.org/abs/2406.04093
> >
> >
> >
> > **Contribution**
> >
> > SAEs improves the interpretability of LLMs by mapping the complex superposition of polysemantic neurons into monosemantic features and composing a sparse dictionary of words. However, traditional performance metrics like MSE and L0 sparsity ignore whether they can acquire interpretable monosemantic features while preserving the semantic relationship of words.
> >
> > In this paper, in complement to traditional MSE or L0 metrics, we propose a suite of evaluations for SAEs to analyze the quality of monosemantic features by focusing on polysemous words. This realizes the holistic evaluation of SAE considering the semantics and interpretablity in addition to compressibility or reconstruction.
> >
> > The qualitative analysis with logit lens in Section 5 demonstrate that a monosemantic SAE feature can correspond to a specific word meaning. Our findings from Section 6 reveal that SAEs developed to improve the MSE-L0 Pareto frontier may confuse interpretability, which does not necessarily enhance the extraction of monosemantic features. In Section 7, the analysis of SAEs with polysemous words can also figure out the internal mechanism of LLMs; deeper layers and the Attention module contribute to distinguishing polysemy in a word.
> >
> > Our semantics-focused evaluation offers new perspective on the polysemy and the existing SAE objective and contributes to the development of more practical and interpretable SAEs.
> >
> >
> > Sincerely,
> > Authors

---

### Public Comment · ~Gábor_Berend1 · 2025-02-10
**Congrats & paper recommendation**

Congratulations to the authors on their work: evaluating SAEs in a way that goes beyond measuring the reconstruction error and the level of sparsity is much needed.
[This paper](https://aclanthology.org/2020.emnlp-main.683/) (disclaimer: authored by me) could also be of potential interest to the authors, as it complements the authors' findings from the perspective of encoder-based models.

---

### Meta-Review · Area_Chair_BnpQ · 2024-12-21

**Metareview:**

The paper proposes a novel framework, PS-Eval, for evaluating Sparse Autoencoders (SAEs) by focusing on their ability to extract monosemantic features from transformer models, particularly in handling polysemous words. This is achieved by repurposing the WiC dataset to assess the features learned by SAEs. The

Reviewers made a lot of suggestions, especially on addressing these conceptual issues, improving the theoretical framework, and clarifying the presentation. I appreciate the author's effort in the rebuttal and am happy to recommend acceptance of the paper.

**Additional Comments On Reviewer Discussion:**

As highlighted in the 'Summary of Discussion', the author offered substantial supplementary information to address the reviewers' concerns.

---

### Decision · Program_Chairs · 2025-01-22

Accept (Poster)